# Frequency-encoded eye tracking smart contact lens for human–machine interaction

Hengtian Zhu[1,7], Huan Yang[1,7], Siqi Xu[2,7], Yuanyuan Ma[3], Shugeng Zhu[1], Zhengyi Mao[1], Weiwei Chen[1,4,5], Zizhong Hu[2], Rongrong Pan[3], Yurui Xu[1,4,5], Yifeng Xiong[1], Ye Chen [1,6] ✉, Yanqing Lu [1] ✉, Xinghai Ning [1,4,5], Dechen Jiang [3,5], Songtao Yuan [2] ✉ & Fei Xu [1,5] ✉

Eye tracking techniques enable high-efficient, natural, and effortless human-machine interaction by detecting users' eye movements and decoding their attention and intentions. Here, a miniature, imperceptible, and biocompatible smart contact lens is proposed for in situ eye tracking and wireless eye-machine interaction. Employing the frequency encoding strategy, the chip-free and battery-free lens successes in detecting eye movement and closure. Using a time-sequential eye tracking algorithm, the lens has a great angular accuracy of <0.5°, which is even less than the vision range of central fovea. Multiple eye-machine interaction applications, such as eye-drawing, Gluttonous Snake game, web interaction, pan-tilt-zoom camera control, and robot vehicle control, are demonstrated on the eye movement model and in vivo rabbit. Furthermore, comprehensive biocompatibility tests are implemented, demonstrating low cytotoxicity and low eye irritation. Thus, the contact lens is expected to enrich approaches of eye tracking techniques and promote the development of human-machine interaction technology.

Recently the rise of wearable flexible devices triggers a revolution in human-machine interaction (HMI)[1–6]. Due to skin-like mechanical property and miniaturized lightweight feature, a seamless and fully connected bridge is built between smart devices and the human body, realizing HMI functions like haptic sensing[7–13], speech recognition[14–16], gesture recognition[17–19], and motion capture[20,21]. Vision accounts for 83% of the information provided when human perceives the outside world[22]. Eye tracking technology, such as the newly released spatial computing device Apple Vision Pro, can analyze intention and cognition through detecting user's attention[23–25], thus enabling high-efficient, natural, and effortless eye-machine interaction[26–30]. Existing eye

tracking devices mostly rely on pupil center corneal reflection technique, which is assisted by a near-infrared light[31–34]. However, this technology is severely limited because of its susceptibility to environmental light interference, the awkward positioning of cameras and light sources, and obstruction caused by user's eyelids and eyelashes, resulting in its poor universality in daily consumer scenarios. Eye tracking technology based on electrooculography (EOG) uses skin electrodes to collect the potential signals of the eye dipole with a positive cornea and negative retina[18,27,35,36], but it is susceptible to interference from muscle electrical signals and has low accuracy. Additionally, it poses a risk to the skin due to the nature of the

[1]National Laboratory of Solid State Microstructures, College of Engineering and Applied Sciences, and Collaborative Innovation Center of Advanced Microstructures, Nanjing University, Nanjing 210023, China. [2]Department of Ophthalmology, The First Affiliated Hospital with Nanjing Medical University, Nanjing 210094, China. [3]The State Key Lab of Analytical Chemistry for Life Science, School of Chemistry and Chemical Engineering, Nanjing University, Nanjing 210093, China. [4]Jiangsu Key Laboratory of Artificial Functional Materials, Nanjing University, Nanjing 210093, China. [5]Chemistry and Biomedicine Innovation Center (ChemBIC), Nanjing University, Nanjing 210093, China. [6]College of Physics, MIIT Key Laboratory of Aerospace Information Materials and Physics, State Key Laboratory of Mechanics and Control for Aerospace Structures, Nanjing University of Aeronautics and Astronautics, Nanjing 211106, China. [7]These authors contributed equally: Hengtian Zhu, Huan Yang, Siqi Xu. ✉e-mail: yechen@nuaa.edu.cn; yqlu@nju.edu.cn; songtaoyuan@njmu.edu.cn; feixu@nju.edu.cn

electrode material. The obvious electrodes also make social interactions awkward. Nowadays, there is an urgent requirement for a wearable and imperceptible eye tracking device to promote the development and application of eye tracking technology in diverse fields, including interactions for individuals with degenerative diseases[37], brain medical diagnosis[38], cognitive science research[39], product human-factor design[40], consumer experience research[41], and driver fatigue detection[42].

Benefitting from the progress of flexible optoelectronic technology, the miniaturized and intelligent contact lenses have been developed[43–45]. Towards augmented reality, smart contact lenses (SCL) are capable of directly projecting images on the retina[46,47]. With their lightweight and implicit design, they are less likely to be perceived by others and are not affected by motion. Towards medical treatment and healthcare, SCL can dynamically monitor physiological changes in real-time, such as intraocular pressure[48–51] and tear glucose levels[52–54]. They can also integrate various medical treatments, including drug delivery[53,55,56], color deficiency correction[57,58], and corneal cell repair[59,60]. The scleral coil-based eye tracking technology, known for its high angular resolution and fast response time, remains the gold standard in eye tracking[61–63]. However, the wired nature of these lenses requires the eye to be anesthetized and is susceptible to slipping during usage, greatly limiting user's acceptance. Moreover, the bulky measurement systems, such as the multiple room-sized generator coil pair, significantly constrain the application scenarios of the eye-tracking SCL.

In this work, a miniature, imperceptible, and wireless eye tracking SCL is proposed for the eye-machine interaction. Using the frequency encoding strategy and advanced spherical conformal preparation technique, the well-designed SCL consists of 4 chip-less passive RF tags with different working frequencies. A portable sweeping-frequency reader is installed at the framed glasses and opposite to the user's eyeball, collecting the tags' signal wirelessly. The changing received signal strengths of multiple tags due to variable coupling coefficients caused by eye movement are employed to track the gazing point and to input the eye commands. Importantly, the proposed SCL has great features including (1) high angular accuracy of eye tracking which is even less than central fovea's vision range; (2) multiple eye-machine interaction modes like the continuous eye-painting, eye-controlled game, web interaction, PTZ camera control, robot vehicle drive, and so on; (3) great comprehensive biocompatibility like low cytotoxicity and low eye irritation. With all these features, the proposed frequency-encoded SCL is expected to enrich the approaches of eye tracking techniques and to renovate the daily interaction modes.

## Results

### Design and characterization of the eye-tracking SCL

Figure 1a demonstrates the wide application potential of the proposed eye tracking SCL. By detecting the gazing direction, the SCL can calculate the real-time gazing point on the virtual screen, enabling the interaction with software, such as giving a like when appreciating Vincent van Gogh's famous artwork *The Starry Night*. Robots also can be eye-controlled through the user-defined eye command input and execute multiple missions like vehicle movement and camera rotation. With miniaturization and portability, the eye tracking system can be used in daily life and bring negligible burden to the user. 4 RF tags, integrated at the peripheral region of SCL, provided the backscattering signal for eye motion detection. The coil-shaped RF tags had different resonant frequency due to well-designed distinct structural parameters (see Supplementary Fig. 1 and Supplementary Table 1), based on the mechanism of resistance-inductance-capacitance (RLC) resonator. The return loss ($S_{11}$) curve, measured by a vector network analyzer (VNA) through a reading coil, provided insights into the response of the tags through wireless detection. The recognition of eye movement and closure was enabled by it. The equivalent impedance $Z_r$ at the terminals of the reading coil is as follow,

$$Z_r = R_r + \sum_{i=1}^{4} j2\pi f L_r \left(1 + \frac{k_i^2 \left(\frac{f}{f_i}\right)^2}{1 + \frac{jf}{f_i Q_i} - \left(\frac{f}{f_i}\right)^2}\right),\tag{1}$$

where $f$ is the excitation frequency. $R_r$ and $L_r$ are the resistance and inductance of the receiver coil respectively. $k_i$ ($i = 1, 2, 3, and\ 4$) is the coupling coefficient between each tag and the receiver coil. And $f_i$ and $Q_i$ are the resonant frequency and quality factor of the RF tags.

A silicone elastomer (MED-6015, NuSil) was employed to encapsulate the tags to isolate the tear environment and enable the spherical morphology. The SCL had the same base arc (8.6 mm) and diameter (13.8 mm) with the commercial contact lens, guaranteeing the fitness with the human cornea (in Fig. 1b). Also, high oxygen permeability and biocompatibility were ensured by the silicone material. To enhance the long-term hydrophilia of the SCL, polyvinylpyrrolidone (PVP) was covered on the surface of the silicone after oxygen plasma treatment[64]. Furthermore, a commercial contact lens (Clariti, Coopervision) was attached to the inner surface of the silicone to provide a safer contact with the cornea. Figure 1c shows the layered schematic diagram of the SCL. The preparation process is described in Methods and Supplementary Fig. 2 in detail.

Comprehensive physicochemical characterizations have been conducted for the SCL. In terms of transparency, there's an unobstructed optical region with a diameter of 4 mm set in the center of the SCL for vision with a high transmission of $89.3 \pm 2.3\ \%$ ($n = 4$) in the visible light range (from 400 to 800 nm), guaranteeing the clear central sight (Supplementary Fig. 3). As shown in Fig. 1d, the static contact angle of the surface of MED-6015 was decreased from 110° to 7° after hydrophilization and maintained a great hydrophilicity within 1 month ($n = 5$). Due to the high capacitance of water, the operation frequency of 4 tags decreased by 2% after hydratation with decreased quality factor. Quality factor can be used to estimate hydration levels and calibrate responses (Supplementary Fig. 4). The SCL had great flexibility and stretchability. Figure 1e shows the photographs of the SCL under normal, compression, and stretching status. To verify the long-term cytotoxicity of the SCL, the viability of cells from human corneas (HCE-T) was measured. Extracts of the SCL were prepared by immersing the lens into the cell culture medium for 24 h. After culturing the cells using extracts with different incubating times (12 h, 24 h, 48 h, and 72 h) and then adding CCK-8 reagent, the absorbance was measured to record the viability of the cells. More details are provided in the Methods. As shown in Fig. 1e, the cell viability using extracts of the SCL remained above 90% after 72 hours of incubating with no notable differences among the groups. This suggests that the SCL was non-cytotoxic over long time wear and would pose little risk of corneal inflammation. The fluorescence images and micrographs of HCE-T after 12-hour culture are demonstrated in Fig. 1f and Supplementary Fig. 5. The cell distribution density and single cell's fluorescence intensity incubated in different extracts were similar, indicating low bio-toxicity. The protein accumulation and disinfection of the SCL were also assessed. A 5 mg/mL of bovine serum albumin-fluorescein conjugate in a phosphate-buffered saline (BSA-FITC, Ruixibio) was employed to mimic protein accumulation from tear. The proteins were incubated on the SCL for 2 h at room temperature, and were removed with a commercial solution (Clear Care; Alcon Laboratories, Inc.). The accumulation of proteins was quantified via fluorescence imaging after protein accumulation and disinfection. Figure 1h presents the accumulation of proteins over the SCL remained lower compared to the bare commercial contact lens with *p < 0.05 after the first disinfection and ***p < 0.001 after the second disinfection using unpaired two-tailed Student's t test ($n = 3$). The lower accumulation of proteins of the SCL is attributed to the hydrophobic nature of MED-6015. Supplementary Fig. 6 presents the representative fluorescence images of the

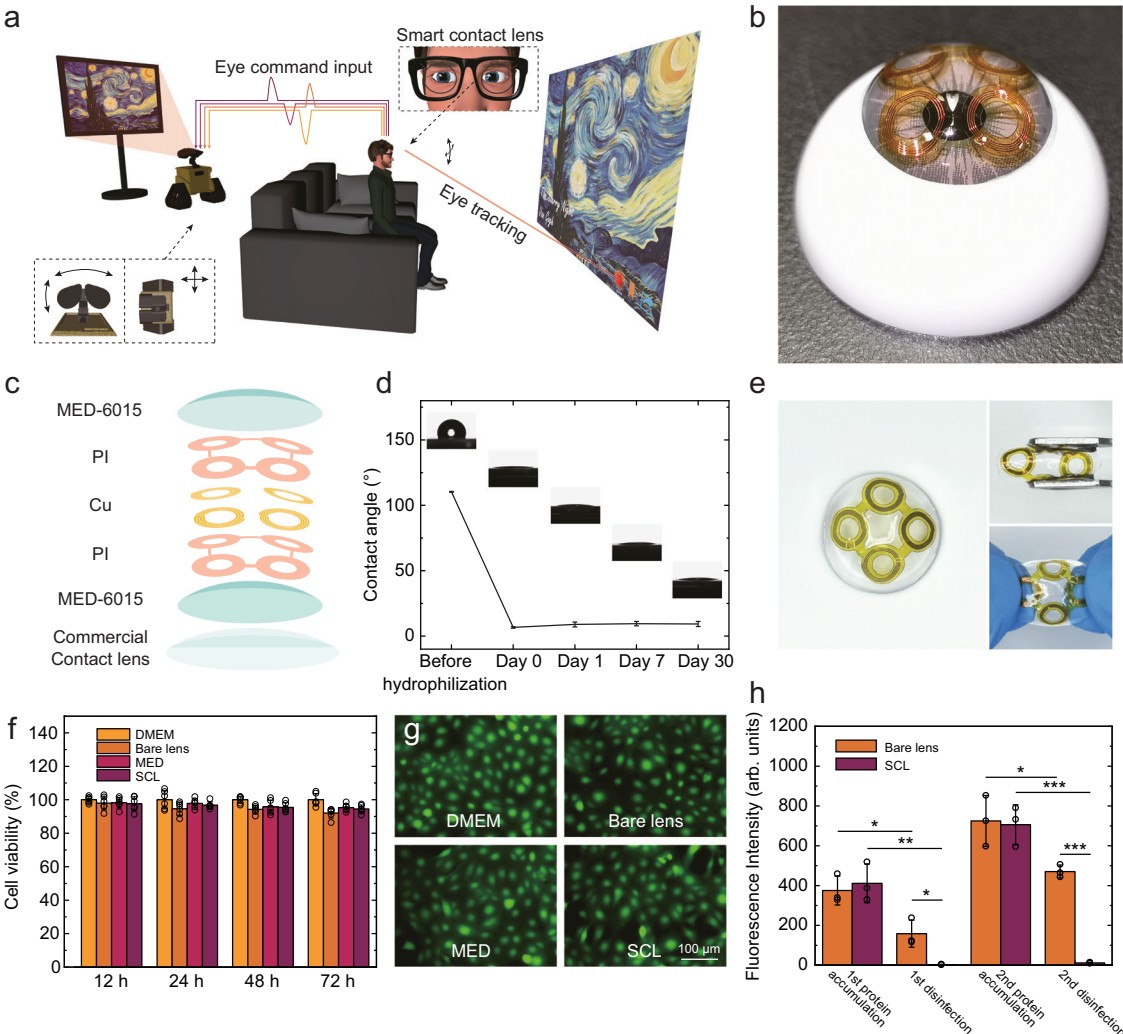

**Fig. 1 | Human-machine interaction by eye-tracking using smart contact lens (SCL). a** Schematic illustration of eye-machine interaction like screen handling when appreciating Vincent van Gogh's famous artwork *The Starry Night* and robot control by eye tracking and eye command emission using SCL. Adapted under terms of the CC-BY license. Copyright 2024, Canino3d, Sketchfab, Inc[67]. **b** Photograph of SCL. **c** Schematic illustration of materials and structures of SCL. **d** Long-term hydrophilia test of SCL. Data are presented as mean with standard deviation of $n = 5$ independent samples. **e** Photographs of SCL under normal, compression, and stretching status. **f** Long-term cytotoxicity test of SCL using human corneal cell lines. Data are presented as mean with standard deviation of $n = 6$ independent cells. **g** Fluorescence images of HCE-T incubated in different extracts. **h** Quantified accumulation of proteins on the bare commercial contact lens (orange column) and the SCL (fuchsia column) after biperiodic protein accumulation and disinfection using unpaired two-tailed Student's t test. $p = 0.0195$ before vs. after first disinfecting the bare commercial contact lens, $n = 3$; $p = 0.00191$ before vs. after first disinfecting the SCL, $n = 3$; $p = 0.0167$ after first disinfecting the bare commercial contact lens vs. the SCL, $n = 3$; $p = 0.0288$ before vs. after second disinfecting the bare commercial contact lens, $n = 3$; $p = 0.000276$ before vs. after second disinfecting the SCL, $n = 3$; $p = 0.0000159$ after second disinfecting the bare commercial contact lens vs. the SCL, $n = 3$. Data are presented as mean with standard deviation of $n = 3$ independent samples. Significant difference was set at ***$p < 0.001$, **$p < 0.01$, and *$p < 0.05$.

commercial contact lens and SCL with the same intensity range, demonstrating better cleaning effect of SCL compared with the commercial contact lens.

## Response model of eye motion

As shown in Fig. 2a, the eye tracking system consists of a frequency-encoded SCL and a portable sweeping-frequency reader installed at the glasses that detects the received signal strengths of the SCL wirelessly. The rotation of the human eyeball can be divided into 2 orthometric directions: pitch rotation and yaw rotation. Because of the multiple working frequency of the peripherally distributed tags, different directional rotations can be distinguished by the SCL. The equivalent electrical circuit is shown in the inset of Fig. 2a. The spatial positions and orientations of multiple tags change when the eyeball rotates, resulting in the variation of coupling coefficient $k_i$ ($i = 1, 2, 3,$ *and 4*), which is manifested as varying amplitude of the

negative resonant signals in the $S_{11}$ curve in Fig. 2b collected by a VNA.

As shown in Supplementary Fig. 7, a 2D eye movement model was built to characterize the response of the SCL. In the model, 2 rotating platforms actuated the pitch rotation and yaw rotation of the model eye which worn the eye tracking SCL, respectively, and a reading coil was placed in front of the model eye. Data was collected from −30° to 30° in the both pitch rotation and yaw rotation. This is comparable to the operation range of other eye tracking devices. Figure 2c shows the response model of SCL for 2D eye movement. The amplitude of each tag was calculated by summing up 20 sampling points (about 16 MHz frequency range) around the corresponding negative resonant signal to reduce the noise of the VNA. Each tag had a maximum signal strength when it faced against the reader at a closest distance. The signal strength gradually decreased when the tag was further away from the opposite coordinate. The

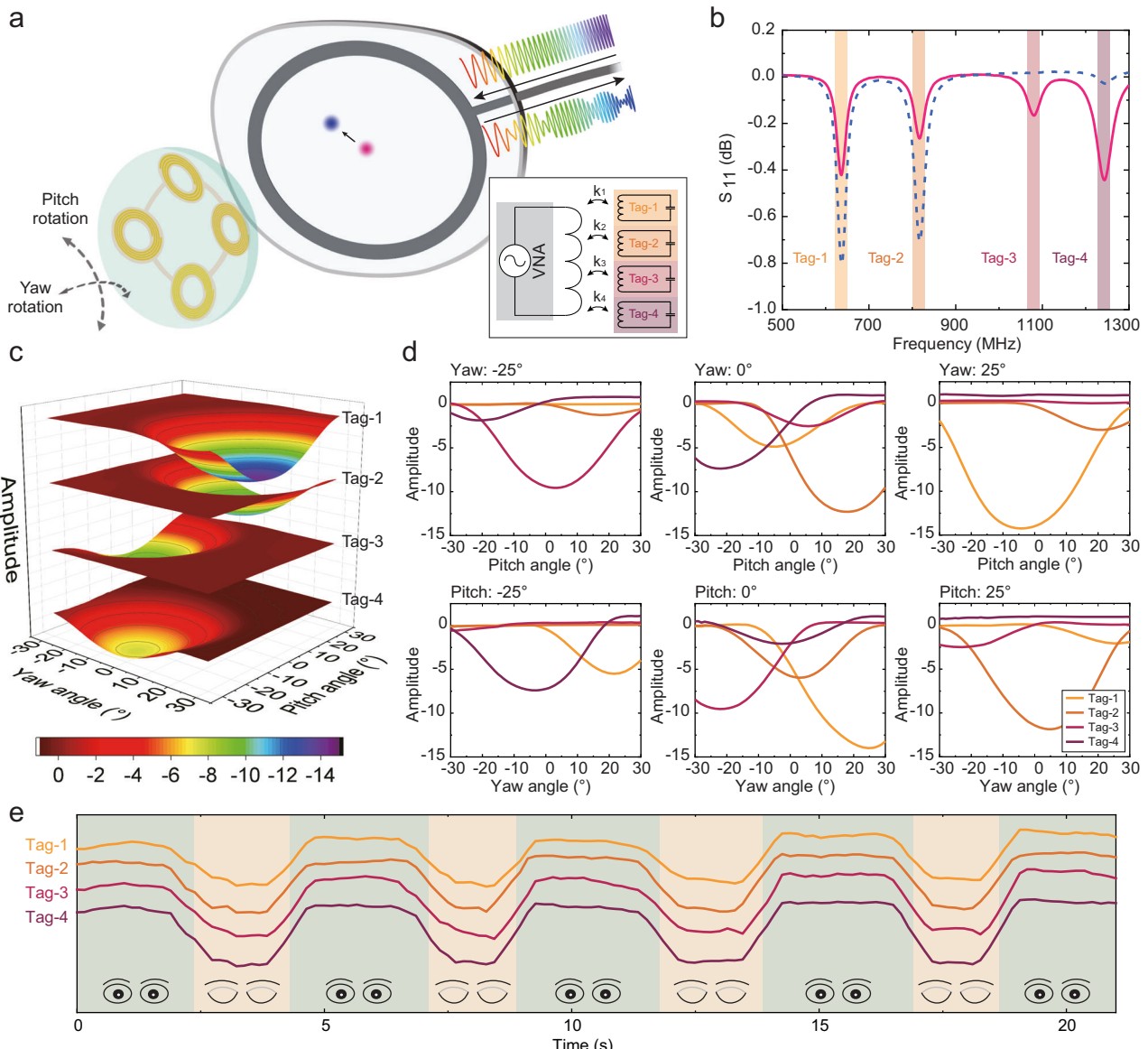

**Fig. 2 | Response model of eye motion. a** Schematic illustration of eye tracking system, including a SCL integrated with 4 frequency-encoded tags and a wide-band sweeping-frequency reader, which consisted of a near-field coil and a vector network analyzer (VNA). Inset: Equivalent circuit of eye tracking system. **b** $S_{11}$ curves with fluctuating tags' intensity caused by varying coupling coefficients $k_i$ under 2 different gazing points. **c** Response model of SCL under 2-dimension eye movement. **d** Details of response model under certain angles. **e** Common-mode eye closure signal caused by eyelid's absorption of electromagnetic field, distinguished from the above differential mode eye-movement signal.

opposite coordinates of tags were different due to the asymmetric structure of the SCL, resulting in the differential mode response to the eye movement. Figure 2d shows the detailed signal under certain angles in the response model. So, the eye movement coordinates can be calculated from the signal amplitude of the 4 tags. 4000-circles pitch rotation and yaw rotation were carried out to evaluate the repeatability of the eye-tracking SCL in Supplementary Fig. 8. The mean standard deviations of 4 tags were 0.061 and 0.043 (with the maximum signal strength of 10.226 and 9.165) respectively in the two tests, demonstrating great repeatability and stability in the 12 hours long tests. Also, the eye-tracking SCL had a great comprehensive robustness for varying environmental light interference and external RF interference. Tags' responses kept consistent when the eye tracking SCL was illuminated by a 24-inch screen (Supplementary Fig. 9) and positioned close to a smartphone in use (Supplementary Fig. 10) and a Wi-Fi router in operation (Supplementary Fig. 11). Slippage of the eye tracking SCL

on the cornea might be an underlying concern for precise eye tracking. Fortunately, the frequency shift of the 4 tags could remind the slippage direction of the SCL and instruct the wear state of the SCL for the user (see Supplementary Fig. 12). The frequency of the one tag increased when touched the edge of the cornea and sclerae of the in vitro porcine eye, and the frequency of the other three tags remained stable when keeping the attachment to the cornea.

What's more, the SCL was capable of detecting eye closure, which is shown in Fig. 2e. The signal amplitude of the 4 tags decreased synchronously when the eyelid closed. It's worth noting that the signal of eye closure was a common-mode signal, which was different from the differential-mode eye movement signal. Additionally, blinking signals could also be identified by the amplitude reduction at frequencies other than the tags' operational frequencies, as shown in Supplementary Fig. 13. So, eye closure and eye movement can be easily distinguished at the same time.

## Precise calligraphy and painting

Accuracy and accessibility are both essential for the eye tracking technique. However, the two features tend to be against each other and are hard to be satisfied in the meantime. Here, we propose a time-sequential eye tracking algorithm based on the response model of the SCL built by an implicit swirling calibration method, realizing high precise eye-calligraphy and painting on a virtual screen in Fig. 3a. The time-sequential eye tracking algorithm began with a calibration procedure, which is shown in Supplementary Fig. 14. The user needed to stare at the swirling pattern on the 27-inch screen, which was assumed to be 60 cm away and opposite to the user. The sight line moved along the 2.5-turn swirling pattern and the signals of multiple tags were recorded with their corresponding coordinates of the gazing points. Then, a thin plate spline interpolation was employed to build the response model of the SCL over the whole screen. The above continuous 2D eye movement model was employed to support the eye movement coordinate calculation. And the calibration process corrects the individual difference in the angle between the geometrical axis and the vision axis (Supplementary Fig. 15) by taking the sight line of the user into consideration other than the geometrical axis of the eyeball. Figure 3b demonstrates the constructed response model and the white lines on the model indicate the swirling pattern in the calibration procedure. The model had a high accuracy, compared with the fingerprint model constructed by traversing over the entire screen region with the step of 1 cm. Supplementary Fig. 16 compares the 2 models and calculates the error spatial distribution of the 4 tags' responses. The standard deviation of the error was about 0.05 with the maximum signal strength was 9.36. The error was mainly located at the border of the screen and the region between the adjacent calibration line of the swirling model. Based on the accurate response model of the SCL, a time-sequential eye tracking algorithm was employed to calculate the gazing point in real time. The algorithm can reduce the threat of the time-varying common-mode drift by calculating the signal difference between adjacent temporal frames. The detailed procedure of the algorithm is expounded in the Supplementary Fig. 14 and Note 1.

The three letters NJU, that mostly covered the whole screen, were calligraphed by eye movement in Fig. 3c. The semitransparent region with 2.1 cm wide around the letters indicates the gazing range of the central fovea, which provides a 2° high-definition view. Figure 3d analyzes the error distribution along X axis and Y axis. In the error statistic, the box is determined by the 25th and 75th percentiles, and the whiskers are determined by the 5th and 95th percentiles. The mean errors were −0.08 cm and 0.30 cm, and the standard deviations were 0.32 cm and 0.28 cm respectively, which were far less than the gazing range. Besides, a one-touch-drawing snake pattern was eye-painting using the eye-tracking SCL in Fig. 3e, which had comparable accuracy with the NJU letters. As shown in Fig. 3f, the mean errors were 0.10 cm and −0.09 cm, and the standard deviations were 0.26 cm and 0.20 cm respectively. Supplementary Fig. 17 shows the eye-drawn NJU letters by using different samples, indicating great reliability of the eye tracking SCL. Using the fingerprint model, the NJU letters and snake pattern obtained a more accurate portrayal in Supplementary Fig. 18. Counting these two patterns together, the standard deviations of the X error and Y error were both 0.21 cm for the fingerprint model. The eye-drawing patterns using the swirling-calibration model had a slightly larger error than using the fingerprint model, as is shown in Supplementary Fig. 19. The error using the fingerprint model was mostly caused by the signal noise from the VNA. More error using swirling-calibration resulted from less accuracy of the constructed response model. Even so, the time-sequential eye tracking algorithm based on the implicit swirling calibration method presented comparable accuracy with the commercial eye tracker and reduced the complexity of the calibration process.

Benefitting from the implicit swirling calibration method, the eye tracking SCL demonstrated a high tolerance in practical applications. As shown in Fig. 3g, due to the asymmetric structure of the SCL, different rotation angles along the axis may occur when wearing the SCL. Using the swirling calibration, the response models were successfully built under different angles. Figure 3h shows the response models under the angles of 30° and 45°, and the NJU letters were both eye-drawn with high accuracy. Through building the response model and recognizing the spatial position of the tags, the torsion of the eyeball, which refers to the rotation about the visual axis and is the third dimension of eyeball rotation, is also able to be measured by our proposed SCL. Besides, the SCL demonstrated strong versatility in accommodating individual differences in corneal curvature. After the establishment of the response model using the implicit calibration method, the SCL was able to accurately eye-draw the NJU letters on the eyeball models with corneal curvatures of 40D, 43D, and 46D (see Supplementary Fig. 20). The mean errors standard deviations in X-axis direction and Y-axis direction were both less than 0.5 cm. The reading distance variation caused by the slippage of the glasses-integrated reader is a critical problem in practical application. The distance between the reader and the eyeball was measured by detecting the self-resonant signal of the reading coil in the $S_{11}$ curve, and then the responses of the tags were modified to match the response model built using the implicit calibration method. The letters NJU were successfully drawn with minimal mean error and standard deviation of less than 0.5 cm in both horizontal and vertical directions, even when drawn in the case of a small distance deviation (see Supplementary Fig. 21).

## Eye-machine interaction by eye command input

The proposed frequency-encoded SCL has been proven to have high angular accuracy owing to the time-sequential eye tracking algorithm based on the implicit swirling calibration method. Precise eye movement detection contributes to recognizing eye commands, which can be user-defined for broader HMI applications like hardware and software control apart from the continuous eye-calligraphy and painting. The eye command is based on the eye movement (up, down, left, and right) and closure. Figure 4a shows the representative tags' signals of the 4 kinds of eye movement. Three kinds of application cases of eye-machine interaction are proposed by eye command input. First, the classic game Gluttonous Snake was played by the eye tracking using our proposed SCL. The moving orientation of the snake was changed according to the eye movement direction. Figure 4b shows the signals of the 4 tags. Based on the differential calculation of the raw signals, 2 signals were obtained in the bottom half of the figure for eye command recognition using threshold judgment. One controlled the up/down motion, and the other controlled the left/right motion. 16 turns were all successfully executed in the 400-second-long game. Figure 4c visualizes the trace of the eye-controlled snake and demonstrates its growth process. Supplementary Movie 1 records the whole process of the Gluttonous Snake game controlled by the eye movement model. Second, the eye-controlled web interaction demo was presented in Fig. 4d-e. In this demo, the webpage switching was realized by the eye commands left and right, that functioned like the hot key Ctrl + Tab and Ctrl + Shift + Tab. The Page Up and Page Down were controlled by the eye commands Up and Down. When the user closed his eye for more than 1 second, Print Screen was carried out to record the web page of the user's interest. Supplementary Movie 2 demonstrates the whole process of the user browsing the webpage and recording the contents of interest using the eye tracking SCL. Except for the interaction with some software, the hardware like the PTZ camera can also be controlled by the SCL using the eye commands input mode. The

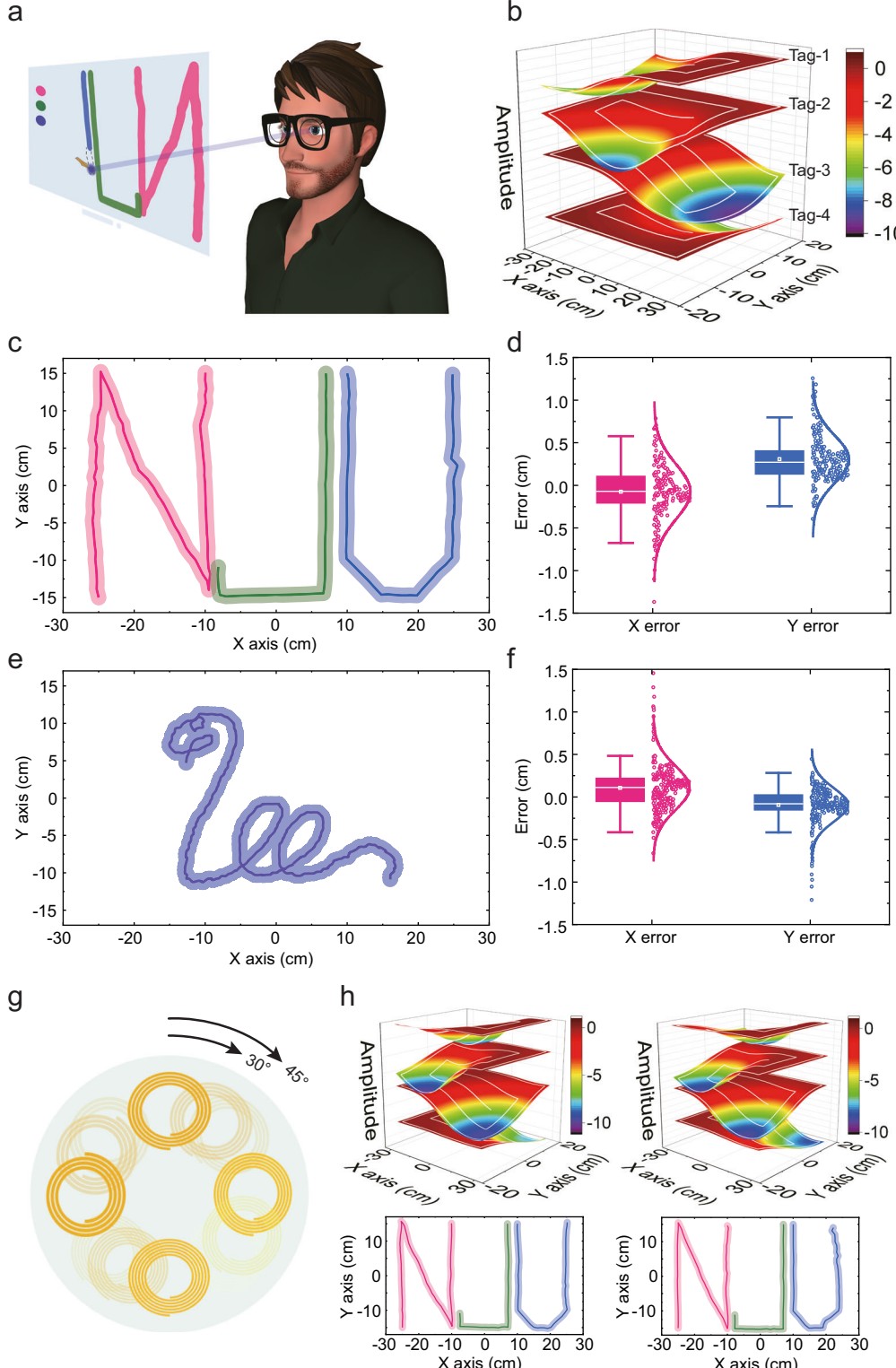

**Fig. 3 | Precise calligraphy and painting by eye tracking. a** Schematic illustration of eye-drawing relying on time-sequential gazing points. Adapted under terms of the CC-BY license. Copyright 2024, Canino3d, Sketchfab, Inc[67]. **b** Constructed eye-movement model using the implicit swirling calibration method. **c–f** Eye-drawing NJU letters and snake pattern with low horizontal and vertical error. The width of the semitransparent trace indicates the gazing range of central fovea. In the error statistic, the middle line is determined by median, the box is determined by the 25th and 75th percentiles, and the whiskers are determined by the 5th and 95th percentiles. $n = 204$ independent gazing points for NJU letters, and $n = 349$ independent gazing points for snake pattern. **g** Schematic illustration of the SCL with different angles when wearing. **h** Constructed eye-movement models and eye-drawing letters under angles of 30° and 45° respectively.

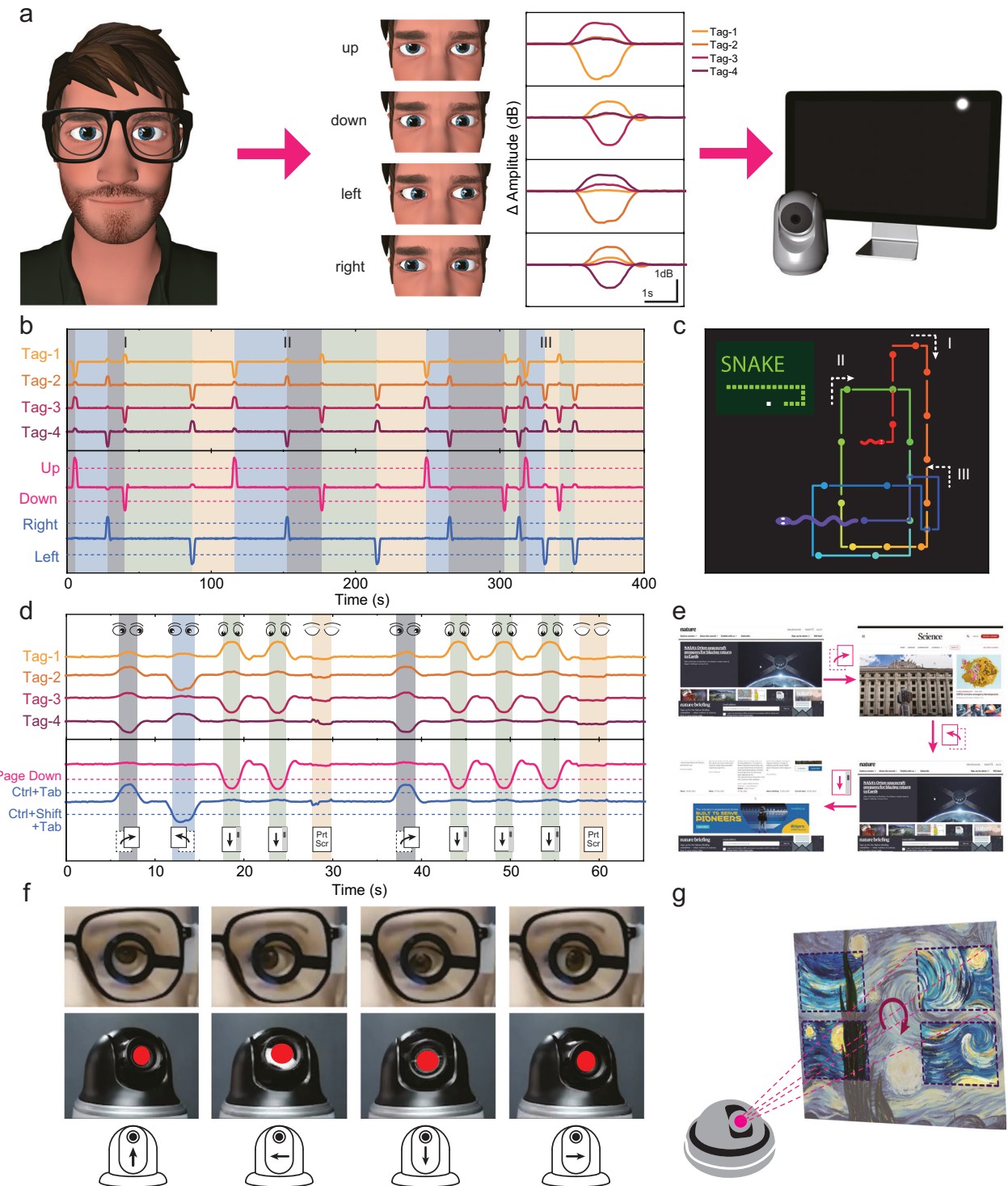

**Fig. 4 | Eye-machine interaction by eye command input. a** Schematic illustration of eye-machine interaction and specific signals of eye commands. Adapted under terms of the CC-BY license. Copyright 2024, Canino3d, Sketchfab, Inc[67]. **b** Raw signals and eye command input when playing the game Gluttonous Snake. Different background colors (blue, green, yellow, and gray) indicate the snake's moving directions (up, down, left, and right). **c** Trace of the eye-controlled snake. **d** Raw signals and eye command input when interacting with webpages. Different background colors (green, blue, gray, and yellow) indicate multiple eye commands (down, left, right, and closure). **e** Eye-controlled webpages. **f** Photographs of the eye movement head model and eye-controlled PTZ camera. **g** Different regions of Vincent van Gogh's artwork *The Starry Night* photographed by the camera.

motion of the camera consists of pan rotation and tilt rotation, which happens to be similar to the eye movement in the 4 directions. Figure 4f-g shows the PTZ camera motion which was naturally controlled by the eye-movement head model and the changing photographic area of the Vincent van Gogh's famous artwork *The Starry Night*. The raw signal of the SCL is shown in Supplementary Fig. 22. Besides, Supplementary Movie 3 demonstrates the motion process of the eye-controlled PTZ camera and the synchronously changing picture. Using the user-defined eye command, the eye tracking SCL was capable of controlling objects both in virtual and actual world in a natural and high-efficiency way.

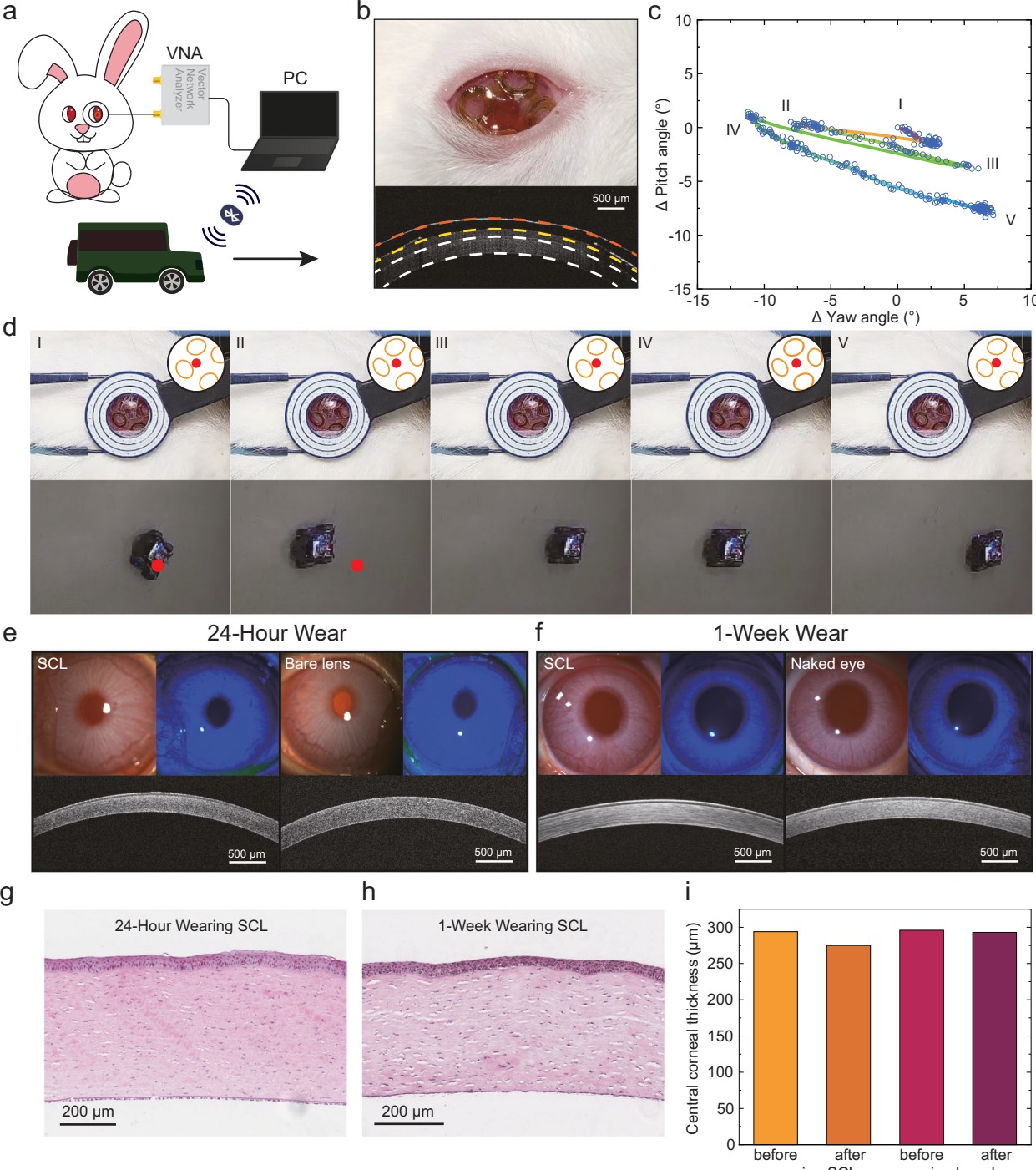

**Fig. 5 | In vivo eye-machine interaction and biocompatibility evaluation.**
**a** Schematic illustration of the rabbit's eye movement controlling the robot vehicle wirelessly. The VNA detected the signal of SCL wireless through the reading coil and transferred to the personal computer (PC), and the robot vehicle was controlled wirelessly through Bluetooth communication by PC. **b** Photograph and OCT image of the rabbit eye with SCL. **c** Trace of the rabbit's eye movement. **d** Photographs of rabbit eye and robot vehicle at specific moments. The insets highlight the location of the reading coil's inner edge and the SCL's 4 tags and the red dots indicate the center of the reading coil. **e** Representative slit lamp photographs, fluorescence images, and OCT images of the rabbit eye after 24-h wear of SCL compared to 24-h wear of bare lens. **f** Representative slit lamp photographs, fluorescence images, and OCT images of the rabbit eye after 8-h daily wear of SCL over 1 week compared to the naked eye. **g, h** Representative micrographs of the rabbit's cornea with H&E staining after 24-h wear of SCL and after 8-h daily wear of SCL over 1 week. **i** Central cornea thickness reduced after 24-h wear of SCL and bare lens due to lens' compression.

## In vivo eye-machine interaction and biocompatibility evaluation

The in-vivo rabbit was employed to verify the function and safety of the eye-tracking SCL. In the in-vivo eye-machine interaction experiment, the eye movement of the rabbit was collected in real-time, and a robot vehicle was controlled wirelessly through Bluetooth communication, as shown in Fig. 5a. Figure 5b represents the photograph and ocular coherence tomography (OCT) image of the rabbit eye wearing the eye-tracking SCL. The eyeball of the rabbit had a similar curvature to the human cornea. So, the SCL was worn conformally on the rabbit eye. The region marked by the white dotted lines in the OCT image

indicates the rabbit's cornea. A commercial contact lens (marked by the yellow dotted line) was placed between the silicon encapsulation (marked by the orange dotted line) and the cornea for more safety. The tags can be observed on the two sides of the OCT image. The motion mode of the robot vehicle (steering and traveling straight) is different from the one of the eye movement. So, the motion of the vehicle was decomposed and a 6-DOF attitude sensor was employed to monitor the motion. The detailed operation procedure is shown in Supplementary Fig. 23. The rabbit's eye rotated back and forth twice within 50 seconds (the raw signals of the tags are in Supplementary Fig. 24). The SCL captured the trace of the eye movement in Fig. 5c. The circle rings in the figure record the eye movement coordinates by frame. The eye movement's velocity can be distinguished by the density of the rings. Figure 5d demonstrates the rabbit eye and the robot vehicle in specific moments. The black circles and yellow ellipses highlight the location of the reading coil's inner edge and the SCL's 4 tags with the red dots showing the center of the reading coil. Subtle eye movement was successfully detected by the SCL and the robot vehicle was driven to the location corresponding to the eye movement coordinates. Supplementary Movie 4 demonstrates the whole process of the robot vehicle controlled by the rabbit eye.

In vivo eye irritation tests were implemented to evaluate the biocompatibility of the eye tracking SCL comprehensively. The eye tracking SCL was worn on one of the rabbit eyes for 24 hours and a commercial contact lens was worn on the contralateral eye for comparison. Figure 5e shows the representative slit lamp photographs, fluorescence images, and OCT images of the rabbit eye after 24-h wear of SCL compared to 24-h wear of bare lens. No corneal injury was observed on the rabbit eye. Images of the rabbit eye before 24-h wear are shown in Supplementary Fig. 25 as a contrast. The representative histopathology images of rabbit cornea with H&E staining after 24-h wear of the SCL and the commercial contact lens are shown in Fig. 5g and Supplementary Fig. 26. There is no notable abnormality like erosion, inflammation, and edema in both images, demonstrating great safety of the SCL, benefitting from the medical grade encapsulation by the silicon elastomer. As shown in Fig. 5i, slight thinning of the rabbit cornea was observed in the OCT images (in Supplementary Fig. 27) after 24-h wear of the SCL and the commercial contact lens, which was caused by the prolonged compression by the lens and eyelid. To further investigate the biocompatibility of the SCL with extended use, long-term in vivo eye irritation tests were performed on rabbit eyes ($n = 3$). The SCL was worn on one of the rabbit's eyes for 8 hours daily over the course of 1 week. Additionally, examination of OCT images (Supplementary Figs. 28–30), slit lamp micrographs (Supplementary Fig. 31-33), and histopathological images (Supplementary Fig. 34) did not reveal any notable abnormalities such as corneal injury, erosion, inflammation, or edema when compared to the naked eye (the representative images shown in Fig. 5f, h). Additionally, the infrared (IR) images in Supplementary Fig. 35 indicate that there was no ocular heating while the SCL was in operation, benefitting from the low port power required for SCL and effective heat dissipation from the ocular tissue and tear fluid.

## Discussion

An eye-tracking SCL based on the frequency-encoded strategy has been proposed for eye-machine interaction. The SCL is capable of monitoring the ocular behavior (eye movement and closure) with 4 well-designed wireless and chip-less RF tags with different operation frequencies embedded in it. The portable sweeping-frequency reader installed at the glasses and opposite to the user's eyeball can collect the tags' signal simultaneously. The whole eye tracking system has a simple structure and light weight for integration with other wearable products, such as the VR head-mount device and AR glasses. Moreover, the eye-tracking SCL owns an ultra-high accuracy of detecting eye movement, using a time-sequential eye tracking algorithm based

on an implicit swirling calibration method. The orientation error is <0.5°, which is even less than the gazing region of the central fovea. In addition, SCL shows excellent robustness to ambient light and common electromagnetic interference, as well as different wearing angles, corneal curvatures and slightly varying reading distances. Two kinds of interaction applications are proposed using the eye tracking SCL. The one is continuous eye-calligraphy and eye-painting on a virtual screen. NJU letters and snake pattern are well portrayed by the continuous eye tracking. The other is the interaction with multiple software/hardware like the gluttonous snake game, web, and PTZ camera, using user-defined eye commands. In addition, the eye tracking SCL has the potential to detect 3D eye movement (including the torsion of the eyeball) and to monitor rapid eye movement in sleep for medical diagnosis. To promote the eye tracking SCL to the practical application, in vivo validation tests and comprehensive biocompatibility tests have been implemented. The in vivo rabbit succeeds in driving the robot vehicle in real time by the eye movement wirelessly. After continuous wear of SCL for 24 hours or daily wear for 8 hours over a week, no abnormality was observed on the rabbit's cornea under the slit-lamp test, histopathology test, and the OCT test. Furthermore, the eye tracking SCL is as safe as the commercial contact lens according to the cytotoxicity test using human corneal cell lines. Supplementary Table 2 summarizes the eye tracking techniques with their advantages and disadvantages. The frequency-encoded SCL, as an innovative wearable eye tracker, enriches the approaches of eye tracking techniques with the advantages of high accuracy, high robustness, and great biocompatibility. Eye-machine interaction, as a natural and efficient interactive mode, has great potential to renovate the operating mode and bring a natural experience to the user based on the wearable eye tracking technique.

In the future, the SCL can be further improved to enhance its practical application for eye tracking. This may involve enhancing flexibility and transparency by using highly-conductive transparent electrodes such as AgNF/AgNW hybrid networks[49,65], or by optimizing the structure of the smart device[48,66]. Collaborative optimization of the SCL, reader, and eye tracking algorithm is crucial in simplifying the calibration process and even achieving calibration-free eye tracking, thus improving accessibility. Additionally, more specialized eye tracking systems can be developed using the smart contact lens and integrating other multiple function modules like field cameras and sensors to achieve intelligent eye tracking applications. These applications could range from consumer behavior research and eye interaction in virtual social settings to diverse medical uses, including visual function assessment, neurological disease diagnosis and treatment, cognitive function assessment, and sleep quality assessment.

## Methods

### Preparation process of eye tracking SCL

The preparation of the SCL includes 3 procedures: preparation of the flexible tags, encapsulation using the medical elastomer, and hydrophilic treatment. As shown in Supplementary Fig. 2, after spin-coating the polyimide (PI) layer (5 μm in thickness) on a clean glass substrate, a 100 nm Cu seed layer was electron-beam deposited on it. A photolithography step was performed to pattern the tags and an electroplating step was performed to thicken the Cu trace of the tags to 8 μm. A thick trace endowed the low resistance to the tags, ensuring the high quality factor of the resonant signals. Then, the second PI layer (5 μm in thickness) was spin-coated to protect the trace from the outer surrounding. A precise laser cutting step was performed to define the PI pattern using a 3-in-1 3D Printer (SM3DP001, Snapmaker). The flexible tags were lift off from the glass after being immersed into the water which reduced the attachment between the PI and glass substrate. Subsequently, the tags were encapsulated by the medical-grade silicone elastomer (MED-6015, NuSil) in a home-made contact lens mold with a temperature of 150 °C for 15 min. The mold provided a base arc

of 8.6 mm and an outer diameter of 13.8 mm to SCL, which was consistent with the commercial contact lens. In order to improve the surface hydrophilia of the SCL, the SCL was treated with oxygen plasma for 180 s, generating reactive groups on the surface. Then, the SCL was immersed in the 22.2% (w/v) PVP (K23-27, Macklin) solution to form polymer brushes. When implementing in vivo rabbit experiments, a commercial contact lens (Clariti, Coopervision) was attached at the inner surface of the SCL to improve the safety further.

## Cytotoxicity tests
Human corneal epithelial cell lines (Cellosaurus CVCL_1272, obtained from Shanghai Zhong Qiao Xin Zhou Biotechnology Co. LTD) were employed to test cytotoxicity. HCE-Ts were maintained in DMEM/F-12 (KGM12500H, KeyGEN BioTECH), supplemented with 10% fetal bovine serum (A5669701, Gibco) and 1% penicillin-streptomycin (complete medium) at 37 °C in a humidified atmosphere of 5% $CO_2$ with medium change every 3 days. After several passages, these cells were harvested and plated at a density of 5000 cells per well. The cells were incubated in complete medium for 24 hours.

Before the experiment, the samples were sterilized with a mixture of ethanol and distilled water (75:25 v/v) for 10 min, rinsed with PBS, and sterilized with ultraviolet irradiation for 12 h. Then, extracts were prepared by immersing both the silicone-embedded tags and bare commercial contact lenses (Clariti, Coopervision) in complete medium at 37 °C for 24 h. In addition, complete medium was used as a control group. Extracts of silicone-embedded tags (MED) and extracts of bare commercial contact lens (Bare lens) were used as the comparisons. The extracts were prepared with contact lens of 0.2 g in complete medium of 1 ml according to ISO 10993-5. A cell viability test ($n = 6$) with different incubating time (12 h, 24 h, 48 h, and 72 h) was performed. The pretreated medium of incubated cells was changed to extracts and was incubated for different durations. The cytotoxicity was assessed by a cell counting kit-8 assay (E1CK-000208, EnoGene). The absorbance was read at 450 nm using a multimode plate reader (Biotek CBM, BioTek). The absorbance values were converted into percentage values relative to the absorbance obtained from only cell growth media. In addition, after 12 h culture, the cells using different extracts was dyed with Calcein AM (C2012, Bryotime) and was incubated for 75 min in dark. Fluorescent images were photoed after rinsing by PBS using a fluorescence microscope (ECLIPSE Ti2, Nikon).

## Eye calligraphy and painting
The eye calligraphy and painting experiments were carried out, using home-made LabVIEW programme. The 2D eye movement model was employed to control the model eyeball, which worn the eye tracking SCL, rotating and gazing at the given coordinates on the virtual screen. The 2D eye movement model consisted of 2 rotating platforms (PRMTZ8, Thorlabs), that had a high resolution of 2 arcsec for eye movement control. The VNA (e5072a ENA, Agilent) was employed to detect the SCL wirelessly through a reading coil. Before eye calligraphy and painting, the response model was constructed by the implicit swirling calibration method. As shown in Supplementary Fig. 14, the eyeball was controlled to gazing along the swirling pattern with a step of 1 cm. Tags' signals and coordinates were collected at each calibration point at the same time. Then, the response model of the SCL for eye tracking was constructed by thin plate interpolation in the Curve Fitting Toolbox of Matlab using the calibration data. When eye calligraphy and painting, the serialized coordinates of NJU letter and snake pattern were predetermined and used as the actual coordinates. Once the eyeball rotated at a targeted coordinate, the signal of the SCL was collected and the measured coordinate was calculated using the time-sequential eye tracking algorithm. The errors along X axis and Y axis were calculated by the difference between the actual coordinate and the measured coordinate. The positive/negative sign (+/-) of the error indicated the direction like right/left or up/down.

## Eye–machine interaction
In the eye-machine interaction experiments, three kinds of application cases like eye-control Gluttonous Snake game, eye-web interaction, and eye-controlled PTZ camera were proposed by eye command input, using home-made LabVIEW programme. The following are the detailed experimental settings and parameters.

**Eye-control Gluttonous Snake game.** The moving orientation of the snake was controlled by the eye command input defined as up, down, left, and right in real time. The eyeball model wearing the SCL was controlled by the 2D rotating platforms. The movement mode was set as rotating 10° in a certain direction and returning back. When playing the game, the VNA (Agilent e5072a ENA) kept reading the signal of the SCL continuously with a sample rate of 7 Hz. The 4 tags' response values were picked out to calculate the 2 differential signals in real time. The one controlled the up/down motion, and the other controlled the left/right motion using threshold judgment. And the thresholds were defined by the differential signals at a deflection of 5° in the 4 directions. User controlled the platforms rotating according to the situation of game. When the signal of the SCL met the preset threshold and lasted for 0.5 s, the corresponding eye movement was identified and eye command was input into the gluttonous snake.

**Eye-web interaction.** The webpage switching was realized by the eye commands left and right, functioned like the hot key Ctrl + Tab and Ctrl + Shift + Tab. The Page Up and Page Down were controlled by the eye command Up and Down. When the user closed his eye for more than 1 second, Print Screen was carried out to record the web page. Similarly, the eye model was controlled by the 2D rotating platforms. The movement mode was set as rotating 10° in a certain direction and then returning back. The thresholds for motion judgement of eye movement were defined by the differential signals at a deflection of 5° in the 4 directions. The eye closure was realized using an artificial eyelid model, which wetted by the PBS to mimic the absorption of tissue, and was recognized via the synchronous drop of the 4 tags' responses. When playing the game, the VNA (Agilent e5072a ENA) kept reading the signal of the SCL continuously with a sample rate of 7 Hz. The user controlled the platforms rotating and the eyelid model to switch, scroll, and record the webpage.

**Eye-controlled PTZ camera.** The motion of the PTZ camera consisted of pan rotation and tilt rotation, which was similar to the eye movement. In the experiment, the camera's motion was controlled by the eye command input defined as up/down and left/right in the manual eye movement model. The maximum rotation angle is limited to 25° in 4 directions. A miniaturized and portable VNA (LibreVNA, ZeenKo) was employed to collect the SCL's signal continuously with a sample rate of 50 Hz. The 4 tags' responses at the maximum angle of the 4 directions were pre-measured to confirm the threshold for motion judgement. When the signal of the SCL met the preset threshold and lasted for 0.5 s, the corresponding eye movement was identified and eye command was transmitted to the PTZ camera wirelessly.

## Robot vehicle drive by rabbit's eye movement
All animal experiments were performed according to the National Institute of Health Guidelines under the protocols approved by the ethics committee at the Nanjing Medical University (SYXK(Su)2021-0023). To verify the effectiveness of the eye tracking SCL, a female New Zealand rabbit (3 kg weight, Jiangsu Qinglongshan Biotechnology Co. LTD), whose eyeball had a similar curvature with the human, was used to drive the robot vehicle via eye movement. The SCL was worn on the one of the rabbit's eyes after local anesthesia, and the rabbit was fixed in a holder that kept its head stable. Then, the reading coil was placed in front of the rabbit's eye, and the portable VNA (LibreVNA, ZeenKo) kept collecting the signal of the SCL continuously with a sample rate of

7 Hz. The eye movement coordinate was calculated in real time. Considering the flexibility and speed of vehicle, the eye movement coordinate was downsampled to 1 Hz. The driving direction and distance were calculated by the eye movement coordinates and transmitted to the vehicle via Bluetooth communication. When the vehicle ran, a 6 DOF attitude sensor (MPU6050, InvenSense) was employed to monitor the steering angle and driving distance.

## Ocular coherence tomography

The anterior segment ocular coherence tomography (AS-OCT) images were acquired during the application of the SCL. The experiment was repeated three times with similar results.

## Biocompatibility tests using in vivo rabbits

All animal experiments were performed according to the National Institute of Health Guidelines under the protocols approved by the ethics committee at the Nanjing Medical University (SYXK(Su)2021-0023). A female New Zealand white rabbit (3 kg weight, Jiangsu Qinglongshan Biotechnology Co. LTD) was used for the 24 h biocompatibility study. The eye-tracking SCL was worn on the left eye while a bare commercial contact lens (Clariti, Coopervision) was worn on the right eye as a comparison. The eyelid was partially sutured to ensure the stable attachment on the cornea. Then the rabbit was housed in a cage and allowed for normal daily activities for 24 h. Multiple ophthalmic examinations including slit lamp biomicroscopy, fluorescein staining, and AS-OCT were performed before and after the 24-h wear of the SCL and the bare commercial contact lens. After completing ophthalmic examinations, the rabbit was humanly euthanized. Its corneas of the both eyes were collected. The collected tissues were routinely fixed, processed in paraffin, sectioned, and stained with H&E for histopathological analysis and reporting. To evaluate the biocompatibility of the SCL during extended wear, 3 female New Zealand white rabbit (3 kg weight, Jiangsu Qinglongshan Biotechnology Co. LTD) were utilized. The SCLs were worn on the left eyes of the rabbits for 8 hours daily over the course of 1 week. The SCLs were disinfected using a commercial contact lens care solution (Clear Care; Alcon Laboratories, Inc.) after each wear. The rabbits' right eyes remained uncovered for comparison. Identical ophthalmic examinations were performed on Day 0 (prior to the first wear), Day 1, Day 2, Day3, and Day 7. After completing week-long examinations, the rabbits were humanly euthanized. Their corneas of both eyes were collected for histopathological analysis and reporting.

## Statistics and reproducibility

Methods for statistical analyses and reproducibility of experiments have been described in detail in previous Methods subsections.

## Reporting summary

Further information on research design is available in the Nature Portfolio Reporting Summary linked to this article.

# Data availability

The authors declare that all data supporting the results in this study are present in the paper and the data sources are uploaded in Supplementary Information with this paper. Any additional requests for information can be directed to, and will be fulfilled by, the corresponding authors. Source data are provided with this paper.

# Code availability

The codes supporting this study's findings are available from https://github.com/yiyinju/Time-sequence-eye-tracking-algorithm.

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

## Acknowledgements

The project was supported by the National Natural Science Foundation of China (61925502,62135007,62305153 and 22025403) and the National Key R&D Program of China (2021YFA1401103).

## Author contributions

All authors provided active and valuable feedback on the manuscript. F.X. and H.Z. initiated the concept and designed the studies; F.X. supervised the work; H.Z., H.Y., and S.Z. led the experiments and collected the overall data; S.X. led the in vivo biocompatibility test; Y.M. led the cytotoxicity test. Z.M. contributed to the preparation of the SCL. W.C., Z.H., R.P., Y.-R.X., Y.-F.X., and Y.C. advised on the experiment. Y.C., Y.L., X.N., D.J., and S.Y. advised on the manuscript. F.X., H.Z., and H.Y. co-wrote the paper.

## Competing interests

The authors declare no competing interests.
