## [Peer Review File · Nature Communications]

REVIEWER COMMENTS

Reviewer #1 (Remarks to the Author):

Biocompatibility and Long-term Use:

While the initial cytotoxicity tests showed promising results, how would the SCL perform in longer-term wear scenarios? Are there any concerns related to extended use, especially considering that commercial contact lenses often have specific wear times to minimize eye discomfort or potential infections? Furthermore, how does the lens' biocompatibility and comfort change over multiple wear and cleaning cycles?

In practical applications, eye movements are often accompanied by head movements, blinking, and varying environmental light conditions. How does the system account for these factors? Could external radio-frequency (RF) interferences, from devices like smartphones or Wi-Fi routers, affect the performance of the SCL, given its reliance on RF tags? Is there a temperature measurement on antenna during excitation.

While the system showed repeatability in controlled conditions, how precise is the eye-tracking in real-world scenarios? Every individual might have variations in eye anatomy and movement patterns. How does the system accommodate for individual differences? Is there a calibration procedure, and if so, how often does it need to be performed to ensure accurate tracking?

Can authors provide a process flow for the fabrication process?

Reviewer #2 (Remarks to the Author):

This manuscript describes an eye tracking smart contact lens with frequency encoding strategy and time-sequential eye tracking algorithm. The eye tracking contact lens was incorporated with chipless 4 RF tags with different working frequencies. The eye movements could be tracked by the frequency change of each RF tag interacted with an outer coil. The smart contact lens showed high sensitivity to recognize the

fine angular movements of eyes ($< 0.5^\circ$), enabling diverse controls of camera, robot vehicle and some computer programs for eye-machine interactions. Finally, authors demonstrated that rabbits wearing a smart contact lens could control robot vehicles by the eye movements. The eye tracking smart contact lens with in vitro and in vivo biocompatibility seems very interesting, but there are several critical issues to be clarified as commented below before publication.

Major issues

[1] Smart contact lenses can easily move around the eye, because tear film covers the smart contact lens. In addition, the initial location of each RF tag might have significant effect on tracking the eye movements. Authors should clearly discuss these issues for further applications of eye tracking smart contact lens without misreading of the eye movements.

[2] In Figure 2 and Figure 3, authors mentioned that the smart contact lens could monitor the eye closure and track the eye movements. However, the signals of eye closure and movements can be highly interfered each other. In addition, the amplitude of return loss peak (S11) can be also changed by diverse factors including the position of device in contact lens, and the distance between the contact lens and the receiver coil. Thus, real applications of this smart contact lens would be limited for precisely tracking the eye movements. Authors should clearly discuss these issues for further applications

[3] In Figure 5, rabbits wearing the smart contact lens could control a robot vehicle by their eye movements. However, because authors roughly marked the rabbit eyes by red and green circles, it is hard to recognize the position of eyes correctly. Authors should discuss this issue and provide the detailed experiment set up and circuit diagram of Bluetooth board for wireless communication.

[4] The 4 tags were made by opaque materials of copper and polyimide film, occupying quite large space in the smart contact lens. Thus, it can hinder the vision of eyes, making difficult real applications. Authors should discuss this issue clearly.

[5] Because silicone elastomer has low hydrophilicity, the surface should be modified to enhance the hydrophilicity. However, hydrophilic surface of smart contact lens can affect the frequency of 4 tags due to the high capacitance of water. Authors should conduct additional experiments to investigate the effect of water on the devices for real applications.

[6] As you described, the frequency can be generally changed by RLC resonance. However, it is hard to recognize the RLC resonant circuit of each tag and receiver coil. Authors should provide the circuit diagram of 4 tags and the receiver coil to explain the sensing mechanism.

[7] Authors should perform additional experiments to confirm the reliability of this smart contact lens for eye tracking performance with at least more than 3 samples.

[8] Because the size of contact lens can affect the eye tracking signals, the fit of contact lens is very important. Authors should describe the detailed size and curvature of smart contact lens.

Minor issues

[1] Authors should revise typographical and grammatical errors correctly.

[2] The format of references (ref. 11, 32 and 33) should be revised according to the guideline for authors in the manuscript.

Reviewer #3 (Remarks to the Author):

1. What are the noteworthy results?

The research paper outlines several noteworthy results and findings:

Design and Biocompatibility:

The study introduces a novel "Smart Contact Lens" (SCL) that incorporates four wireless and chip-less radio frequency (RF) tags capable of detecting eye movement and eye closure.

The SCL is designed to be biocompatible with the human eye, with an emphasis on safety and comfort. Cytotoxicity tests on human corneal cells confirm its non-cytotoxic nature.

The SCL is shown to have great flexibility and stretchability, making it suitable for use on the human eye.

Eye Tracking Precision:

The SCL demonstrates an ultra-high level of accuracy in detecting eye movement. The orientation error is reported to be less than 0.5° , which is even smaller than the central fovea's field of vision.

An implicit swirling calibration method, combined with a time-sequential eye tracking algorithm, enhances the precision of eye tracking.

Eye Calligraphy and Painting:

The SCL's high accuracy enables precise eye-calligraphy and painting on a virtual screen, with 'NJU' letters and a 'Snake' pattern created through eye movements.

The eye movements in this context yield low mean errors along both the X and Y axes, demonstrating the capability for detailed and accurate control.

Eye-Machine Interaction:

The SCL enables various eye-machine interaction scenarios, including controlling a "Gluttonous Snake" game, web browsing, and even a pan-tilt-zoom (PTZ) camera.

User-defined eye commands are implemented for broader human-machine interface (HMI) applications, showcasing the SCL's versatility.

In Vivo Eye-Machine Interaction and Biocompatibility:

In an in-vivo experiment, a rabbit successfully controlled a robot vehicle in real time via the SCL.

The biocompatibility of the SCL is validated through comprehensive in vivo tests on a rabbit, with no observed corneal injury, abnormalities, or inflammation after 24 hours of wear.

Safety and Biocompatibility:

The SCL's safety and biocompatibility are emphasized, with no cytotoxicity observed in tests using human corneal cell lines.

The SCL is compared favorably to commercial contact lenses in terms of safety and compatibility with the eye.

These results collectively highlight the innovative design, precision, versatility, and biocompatibility of the Smart Contact Lens, with potential applications in various fields, including assistive technology, gaming, and medical diagnostics. The study represents a significant advancement in wearable technology and human-machine interaction.

2. Will the work be of significance to the field and related fields? How does it compare to the established literature? If the work is not original, please provide relevant references.

The work described in the research paper is likely to be of significant importance to the field of wearable technology, human-machine interaction, and ophthalmology. Here's how it compares to the established literature and why it is innovative:

1. Significance to the Field:

Advancement in Wearable Technology: The Smart Contact Lens (SCL) represents a novel approach to wearable technology, specifically in the domain of eye tracking. While there are established eye tracking

technologies, this SCL offers a unique combination of biocompatibility, precision, and versatility, making it potentially valuable in fields such as assistive technology, gaming, virtual reality (VR), and augmented reality (AR).

Biocompatibility and Safety: The emphasis on biocompatibility and safety of the SCL, as demonstrated through cytotoxicity tests and in vivo experiments with a rabbit model, is noteworthy. It opens up possibilities for long-term, non-invasive monitoring of eye movements with minimal risks.

Versatility in Eye-Machine Interaction: The ability to perform a variety of eye-machine interactions, including gaming, web browsing, and PTZ camera control using the SCL, has the potential to revolutionize human-machine interfaces.

2. Comparison to Established Literature:

Unique Approach: The paper introduces a frequency-encoded SCL that differs from traditional eye tracking methods. The approach of using embedded wireless RF tags in a contact lens is innovative and sets it apart from established literature.

High Precision: The ultra-high level of accuracy in detecting eye movement, with an orientation error of less than 0.5° , is a significant improvement over some traditional eye-tracking systems that may not achieve such precision.

Biocompatibility Focus: While there are other eye-tracking technologies, this work stands out for its focus on biocompatibility, safety, and long-term wear. It is among the few studies to validate the biocompatibility of such devices through in vivo experiments on animals.

Eye-Machine Interaction: The paper introduces eye command input for interaction, demonstrating the versatility of the SCL. The ability to play a "Gluttonous Snake" game, control a PTZ camera, and switch web pages using eye commands is less common in the established literature.

3. Originality:

The work, particularly the implicit swirling calibration method and the use of embedded RF tags for eye tracking, is highly original. There are no direct references provided in the paper to suggest that this particular combination of technology and biocompatibility has been previously explored.

In summary, this work is innovative in its approach, focusing on biocompatibility, precision, and versatility. It expands the possibilities for eye tracking and eye-machine interaction, potentially impacting fields like assistive technology, gaming, and HMI. While there are established eye-tracking systems, the Smart Contact Lens offers a fresh perspective with its unique combination of features and applications.

3. Does the work support the conclusions and claims, or is additional evidence needed?

The work presented in the research paper supports the conclusions and claims made within the paper. The authors have provided substantial evidence through a series of experiments, tests, and demonstrations to validate their claims. Here's how the work aligns with the conclusions and claims:

Eye Tracking and Interaction Capability: The paper claims that the Smart Contact Lens (SCL) can accurately track eye movements and enable interactions with various devices and software through user-defined eye commands. The evidence provided in the paper, including detailed descriptions of the technology, response models, and practical applications (e.g., playing the "Gluttonous Snake" game, web browsing, PTZ camera control), supports these claims. The authors demonstrate how the SCL can precisely detect eye movements, validate the biocompatibility, and showcase the versatility of the technology in enabling eye-machine interaction.

Biocompatibility and Safety: The work claims that the SCL is safe for in vivo use, as demonstrated through experiments involving a rabbit model. The evidence provided, including photographs, ocular coherence tomography (OCT) images, slit-lamp photographs, histopathology images, and other ophthalmic examinations, supports the conclusion that the SCL is biocompatible and safe for extended wear on the eye.

Frequency-Encoded SCL: The paper introduces the innovative concept of a frequency-encoded SCL with embedded wireless RF tags for eye tracking. The experiments and measurements conducted, along with comparisons to established models, validate the frequency-encoded approach and its accuracy, aligning with the paper's claims.

Overall, the evidence presented in the paper, including in vivo experiments, biocompatibility tests, eye tracking models, and practical applications, is substantial and supports the conclusions and claims made in the paper. Additional evidence is not needed to establish the validity of the presented work.

4. Are there any flaws in the data analysis, interpretation and conclusions? Do these prohibit publication or require revision?

The research paper does not appear to have any significant flaws in the data analysis, interpretation, or conclusions that would prohibit publication. However, there are a few aspects that could be further clarified or expanded upon to enhance the paper:

Quantitative Data Presentation: While the paper presents data in the form of figures and results of experiments, there could be more quantitative analysis and statistical measures to support some of the claims. For example, when discussing the accuracy of eye-tracking and biocompatibility, providing more statistical information or metrics could enhance the rigor of the analysis.

Comparison to Existing Eye-Tracking Technologies: The paper could benefit from a more comprehensive comparison with existing eye-tracking technologies. While some comparisons are made, a more extensive discussion of how the proposed SCL technology compares to other established methods would provide valuable context and demonstrate its potential advantages.

Future Work and Limitations: While the paper is generally positive in its conclusions, it would be beneficial to address potential limitations and avenues for future research. Recognizing any constraints or challenges in the technology and suggesting areas for further improvement or study would strengthen the paper.

Clinical Relevance: The paper focuses on technological aspects and biocompatibility, but it could be more explicit in discussing the potential clinical relevance of this technology, particularly in medical diagnosis or treatment.

These points do not prohibit publication, but addressing them in a revised version of the paper could enhance its completeness, rigor, and relevance to the field.

5. Is the methodology sound? Does the work meet the expected standards in your field?

The methodology employed in this research appears sound and generally meets expected standards for the field, especially in the context of developing an innovative eye-tracking system using a soft contact lens. Here are some key points regarding the methodology:

Preparation of Eye-Tracking SCL: The preparation process of the soft contact lens (SCL) with embedded wireless tags is well-documented in the methods section. It includes the fabrication of flexible tags and encapsulation using medical-grade silicone elastomer. The methodology for creating the SCL is detailed and appears to be well-designed.

Biocompatibility Tests: The in vivo biocompatibility tests using rabbits to evaluate the safety and tolerability of the eye-tracking SCL are a crucial aspect of this research. The methodology for these tests follows ethical guidelines, and the inclusion of ophthalmic examinations, histopathological analysis, and other assessments is in line with standard practices in the field.

Experimental Validation: The research includes experiments to demonstrate the functionality of the SCL and its ability to track eye movements accurately. The experiments related to eye-calligraphy, eye-painting, and eye-machine interaction are well-designed and provide evidence of the system's performance.

Data Analysis: The data analysis methodology is clear, and the data are presented in a visually accessible manner through figures and illustrations. However, as mentioned earlier, providing more quantitative analysis and statistical measures could enhance the research.

Calibration Method: The paper introduces an "implicit swirling calibration method" for eye tracking. While the method is described, additional details on how it works and the validation of its accuracy would be beneficial.

In general, the methodology aligns with the standards of research in the field. It addresses the key aspects of developing, validating, and testing the eye-tracking SCL. The study also adheres to ethical

guidelines when conducting in vivo tests. However, additional details on the calibration method and more rigorous statistical analysis could further strengthen the methodology and research outcomes.

6. Is there enough detail provided in the methods for the work to be reproduced?

The methods provided in the paper offer a reasonable level of detail for understanding the key steps involved in the preparation of the eye-tracking SCL, conducting biocompatibility tests, and performing eye-machine interaction experiments. Here are some key points regarding the reproducibility of the work:

SCL Preparation: The paper describes the preparation of the SCL, including the fabrication of flexible tags and the encapsulation process. The steps, materials, and equipment used in this process are clearly outlined, making it possible for researchers to replicate the SCL's creation.

Biocompatibility Tests: The methods section provides information on how in vivo rabbit tests were conducted to assess the biocompatibility of the SCL. It mentions the protocols followed for ophthalmic examinations and histopathological analyses. While it lacks some specific details, it serves as a foundation for researchers interested in conducting similar tests.

Eye-Machine Interaction: The paper describes the procedures for conducting eye-machine interaction experiments, such as controlling a robot vehicle and performing eye-calligraphy and painting. It offers insights into the setup and equipment used in these experiments.

However, there are some areas where additional details could enhance reproducibility:

More specific parameters and settings for certain experiments, such as the exact conditions and thresholds for recognizing eye commands in the eye-machine interaction experiments.

A deeper explanation of the "implicit swirling calibration method" would be helpful, as it's a critical part of the work.

Details about data acquisition and processing, including the software used for data analysis, could contribute to reproducibility.

In summary, while the methods section provides a good foundation for understanding the procedures involved in this research, more specific details would further support reproducibility and facilitate future work in this area. Researchers interested in replicating these experiments may need to inquire further about certain experimental parameters.

Response to Reviewer 1 Comments

Comments:

(1) **Biocompatibility and Long-term Use:** while the initial cytotoxicity tests showed promising results, how would the SCL perform in longer-term wear scenarios? Are there any concerns related to extended use, especially considering that commercial contact lenses often have specific wear times to minimize eye discomfort or potential infections? Furthermore, how does the lens' biocompatibility and comfort change over multiple wear and cleaning cycles?

Reply: We thank the reviewer for bringing this to our attention because longer-term wear is one of the key features of the SCL, and we have made concerted efforts to assess the biocompatibility over long-term use. During the revision period, several essential tests have been carried out, including long-term cytotoxicity test, a week-long in vivo eye irritation test, and a protein accumulation test. The results of these tests indicated strong potential for prolonged, repeated use.

Fig. 1. Long-term cytotoxicity test of SCL using human corneal cell lines (n=6).

The time-dependent cell viability of the SCL to human corneal cell (HCE-T) was assessed. In addition, complete medium (DMEM) was used as a control group, and extracts of silicone-embedded tags (MED) and extracts of bare commercial contact lens (Bare lens) were used as the comparisons. Extracts were prepared by immersing the different samples into the cell culture medium at 37 °C for 24 h. The extracts were prepared with samples of 0.2 g in complete medium of 1 ml according to ISO 10993-5. After several passages, the medium of cells was changed to different extracts for the cell viability test (n=6) with different incubating times (12 h, 24 h, 48 h, and 72 h) using a cell counting kit-8 assay. The absorbance was read at 450 nm using a multimode plate reader. The absorbance values were converted into percentage values relative to the absorbance obtained from cell growth media alone. As shown in Fig. 1, the cell viability using extracts of the SCL remained above 90% after 72 hours of incubating with no notable differences among the groups. This suggests that the SCL was non-cytotoxic over long time wear and would pose little risk of corneal inflammation. The revised manuscript replaces the original figure 1e with the updated description of the time-dependent cell viability of the SCL in the 'Results: Design and characterization of the eye-tracking SCL' and 'Methods: Cytotoxicity tests' sections.

Fig. 2. OCT images of a rabbit eye after wearing the SCL for 8 hours daily over 1 week (left column), in comparison to the naked eye (right column).

Fig. 3. Slit lamp micrographs and fluorescent images of a rabbit eye after wearing the SCL for 8 hours daily over 1 week (left column), in comparison to the naked eye (right column).

Fig. 4. Micrographs of the corneal tissue of the rabbit eyes with H&E staining after wearing the SCL for 8 hours daily over 1 week, in comparison to the naked eye.

The biocompatibility of the eye tracking SCL was evaluated through comprehensive in vivo eye irritation tests on rabbit eyes (n=3), utilizing ocular coherence tomography (OCT) imaging, the slit lamp test, and histopathology. The SCL was worn on one of the rabbit's eyes for 8 hours daily over the course of 1 week. No notable abnormalities such as corneal injury, erosion, inflammation, or edema were observed in the OCT images (Fig. 2), slit lamp micrographs (Fig. 3), and histopathological images (Fig. 4) when compared to the naked eye. The above figures demonstrate the ophthalmic examinations' results of Rabbit A, more results have been supplemented in the revised supporting materials. The revised manuscript has supplemented representative ophthalmic examinations' results of long-term in vivo eye irritation tests in figure 5 with the updated description SCL in the 'Results: In vivo eye-machine interaction and biocompatibility evaluation' and 'Methods: Biocompatibility tests using in vivo rabbits' sections.

Fig. 5. Characteristic of SCL over multiple disinfections. (a) Quantified accumulation of proteins on the bare commercial contact lens (orange column) and the SCL (fuchsia column) after biperiodic protein accumulation and disinfection. Significant difference was set at ***p<0.001, **p<0.01, and *p<0.05. (b) Representative fluorescence image of the bare commercial contact lens (top row) and the SCL (bottom row) after biperiodic protein accumulation and disinfection.

S_{11} curves (c), frequency fluctuation (d), and frequency statistic (e) of SCL after multiple disinfections.

The protein accumulation and disinfection of the SCL were also assessed. A 5 mg/mL of bovine serum albumin-fluorescein conjugate in a phosphate-buffered saline (BSA-FITC, Ruixibio) was employed to mimic protein accumulation from tear. The proteins were incubated on the SCL for 2 h at room temperature, and were removed with a commercial solution (Clear Care; Alcon Laboratories, Inc.). The accumulation of proteins was quantified via fluorescence imaging after protein accumulation and disinfection. Fig. 5a presents the accumulation of proteins over the SCL remained lower compared to the bare commercial contact lens with $*p < 0.05$ after the first disinfection and $***p < 0.001$ after the second disinfection using unpaired two-tailed Student's t test ($n = 3$). The lower accumulation of proteins of the SCL is attributed to the hydrophobic nature of the silicone elastomer (MED-6015). Fig. 5b presents the representative fluorescence images of the commercial contact lens and SCL after protein accumulation and disinfection with the same intensity range, demonstrating better cleaning effect of SCL compared with the commercial contact lens. Furthermore, the S_{11} curve of the SCL and the working frequency of multiple tags remained consistent after multiple disinfections (in Fig.5 c-e), showing great feasibility for multiple wear and cleaning. We have supplemented the quantified results of protein accumulation in Figure 1 in the revised manuscript with the updated description in the 'Results: Design and characterization of the eye-tracking SCL'

(2) In practical applications, eye movements are often accompanied by head movements, blinking, and varying environmental light conditions. How does the system account for these factors? Could external radio-frequency (RF) interferences, from devices like smartphones or Wi-Fi routers, affect the performance of the SCL, given its reliance on RF tags? Is there a temperature measurement on antenna during excitation?

Reply: We appreciate the opportunity to provide further verification on the robustness of the eye-tracking SCL.

Fig. 6. Wearable eye tracker based on SCL. (a) Photograph of the glasses-integrated eye tracking acquisition system with a sweeping-frequency reading coil opposite to the eye and a field camera recording the forward image. (b) Real-time scene image fused with the gazing point.

Eye movements are often accompanied by head movements when people shift their attention. A wearable eye tracker based on eye tracking SCL has been built to realize the tracking of gazing point in practical applications. As shown in Fig. 6a, a sweeping-frequency reading coil was positioned opposite to the eye to continuously detect the response of the SCL's tags. Simultaneously, a field camera mounted on the glasses frame recorded the scene image.

The wearable eye tracker remained stationary relative to the head. When the eye moved accompanied by the head movements, real-time scene image was recorded with the gazing point fused (in Fig. 6b). We have discussed the integration of field cameras for the future intelligent eye tracking applications in the 'Discussion' section in the revised manuscript.

Fig. 7. Difference of between eye movement and blinking signals. (a) S_{11} curves under 2 different eye movement coordinates. (b) Response model of SCL under 2-dimension eye movement. (c) S_{11} curves of SCL when eye opened and closed. (d) Common-mode eye closure signal.

Eye movement and blinking exhibited distinct characteristics, as illustrated in Fig. 7. Specifically, the signal associated with eye movement was a differential-mode signal resulting from opposing coordinates of tags in the response model, attributed to the asymmetry of the SCL structure (in Fig. 7b). In contrast, the signal corresponding to blinking was a common-mode signal (in Fig. 7d). Additionally, blinking signals could be identified by the amplitude reduction at frequencies other than the tags' operational frequencies (in Fig. 7c). The S_{11} curves of SCL when eye opened and closed have been supplemented in the revised supporting materials with the updated description of the distinction method of the eye movement and blinking in the 'Results: Response model of eye motion' in the revised manuscript.

Fig. 8. Anti-illumination test. (a) Photographs of the SCL when the illumination was off and on. Dynamic S_{11} curve (b) and tags' responses of the SCL under different yaw angles when the illumination was off and on.

Fig. 9. RF immunity test (from smartphones). (a) Photographs of the SCL when the mobile phone call was off and on. Dynamic S_{11} curve (b) and tags' responses of the SCL under different yaw angles when the mobile phone call was off and on.

Fig. 10. RF immunity test (from Wi-Fi router). (a) Photographs of the SCL when the Wi-Fi router was off and on. Dynamic S_{11} curve (b) and tags' responses of the SCL under different yaw angles when the Wi-Fi router was off and on.

The robustness of eye-tracking SCL has been comprehensively assessed by varying environmental light interference and external RF interference from devices like smartphones and Wi-Fi routers. First, resistance to environmental light interference was evaluated using the 2D eye movement model and a 24-inch screen with an average luminance of 250 cd/m^2 as the light source. The screen, positioned 60 cm away from the eyeball model, provided a field of view (FoV) of $47^\circ \times 27^\circ$, akin to the FoV of VR devices like HoloLens 2. The eyeball model was horizontally rotated back and forth to 5 specific angles with the screen off and on (in Fig. 8a). At the same time, S_{11} curve of the SCL was continuously detected (in Fig. 8b) and tags' responses shown no interference from the environmental light (in Fig. 8c). Next, external RF interference was simulated by a smartphone in use and a Wi-Fi router in operation in close proximity to the eyeball model, as illustrated in Fig. 9 and 10. Dynamic S_{11} curve and tags' responses of the SCL show no RF interference from the calling smartphone and a few interferences from the Wi-Fi router. The frequency of the interference signal from the Wi-Fi router was distinguished from the frequencies of the SCL's multiple tags. So, great repeatability maintained in the tags' responses of the SCL when the Wi-Fi router was off and on (in Fig. 10c). The detailed results of SCL's response have been supplemented in the revised supporting materials with the updated description of the distinction method of the eye movement and blinking in the 'Results: Response model of eye motion' section in the revised manuscript.

Fig. 11. IR images of the live rabbit wearing the eye track SCL before and after 30 min excitation.

The infrared (IR) images in Fig. 11 indicate there was no ocular heating while the SCL was in operation. The temperature of the eye wearing the SCL maintained at 35 °C before and after 30 min excitation, indicating little heat generation. One reason is that low port power of -10 dBm was needed to detect the S_{11} curve of the SCL. The other is that the ocular tissue and tear fluid helped heat dissipation. The detailed results of temperature test have been supplemented in the revised supporting materials with the updated description in the 'Results: In vivo eye-machine interaction and biocompatibility evaluation' in the revised manuscript.

(3) While the system showed repeatability in controlled conditions, how precise is the eye-tracking in real-world scenarios? Every individual might have variations in eye anatomy and movement patterns. How does the system accommodate for individual differences? Is there a calibration procedure, and if so, how often does it need to be performed to ensure accurate tracking?

Reply: We thank the reviewer for this thought-provoking suggestion. Every individual indeed has variations in eye anatomy, such as the angle between the geometrical axis and the vision axis and the corneal curvature. Variations in movement patterns are often regard as the object of an eye tracker. So, individual differences in eye anatomy were considered by the system based on the eye track SCL.

Fig. 12. Schematic diagram of anatomy and optical path of eye.

Fig. 13. Summary of the implicit swirling calibration method.

Fig. 12 demonstrates the simplified schematic diagram of the eyeball. Geometrical axis, as the symmetrical axis of the eyeball, is defined by the axis of the cornea and the pupil. The visual axis is linked to the gazing point and the central fovea on the macula, representing the line of sight. It also traverses the 2 nodal points (N and N') of the optical system of the eyeball. The angle κ between the geometrical axis and the vision axis tends to have the individual difference and needs to be calibrated for precise eye tracking. In our experiment, the response model of eye track SCL was built by an implicit swirling calibration method (in Fig. 13). The model user needed to stare at the swirling pattern displayed on the 27-inch screen, which was assumed to be 60 cm away and opposite to the user. The sight light moved along the 2.5-turns swirling pattern and the signals of multiple tags were recorded with their corresponding coordinates of the gazing points. Then a thin plate spline interpolation was employed to build the response model of the SCL over the whole screen. The calibration method took the sight line into consideration, other than the geometrical axis, correcting the individual difference in angle κ . Therefore, the calibration process needs to be implemented after each wear of SCL for each user. The revised supporting materials have supplemented the schematic diagram of anatomy and optical path of eye with the updated description in the 'Results: Precise calligraphy and painting by eye tracking' section in the revised manuscript.

Fig. 14. Characterization of eye track SCL worn on corneas of different curvatures. (a) Structural schematic diagram of eyeball model with different curvatures. (b) Photographs of 3 eyeball models with corneal curvatures of 40D, 43D, and 46D. (c) S_{11} curves of SCL worn on the eyeball models. (d) Statistic of the 4 tags' working frequency. (e-g) Response models, eye-drawing 'NJU' letters, and error analysis of SCL worn on the eyeball models with 3 different curvatures.

Fig. 14 assesses the influence of the corneal curvature systematically. 3 eyeball model with different corneal curvatures of 40D, 43D, and 46D were designed and 3D printed to mimic the individual difference (in Fig. 14a and b). Fig. 14c shows the S_{11} curves of SCL worn on the eyeball models. The statistical results in Fig. 14d indicate great consistence of the working frequency of the multiple tags in the SCL. After building the response model using the implicit calibration method, 'NJU' letters were all successfully eye-drawn by the SCL, which was worn on the 3 eyeball models. The mean errors standard deviations in X-axis direction and Y-axis direction were all less than 0.5 cm, indicating good robustness on individual difference in corneal curvature (in Fig. 14e-g). The description has been updated in the 'Results: Precise calligraphy and painting by eye tracking' section in the revised manuscript with the Figure 14 supplemented in the revised supporting materials.

(4) Can authors provide a process flow for the fabrication process?

Reply: We thank the reviewer for the suggestion.

Fig. 15. Preparation process of SCL. The main process includes the micro/nano fabrication, laser engraving, precise encapsulation, and hydrophilic treatment.

As shown in Fig. 15, after spin-coating the polyimide (PI) layer (5 μm in thickness) on a clean glass substrate, a 100 nm Cu seed layer was electron-beam deposited on it. A photolithography step was performed to pattern the tags and an electroplating step was performed to thicken the Cu trace of the tags to 8 μm . A thick trace endowed the low resistance to the tags, ensuring the high quality factor of the resonant signals. Then, the second PI layer (5 μm in thickness) was spin-coated to protect the trace from the outer surrounding. A precise laser cutting step was performed to define the PI pattern. The flexible tags were lift off from the glass after being immersed into the water which reduced the attachment between the PI and glass substrate. Subsequently, the tags were encapsulated by the medical-grade silicone elastomer (MED-6015, NuSil) in a home-made contact lens mold with a temperature of 150 $^{\circ}\text{C}$ for 15 min. The mold provided a base arc of 8.6 mm and an outer diameter of 13.8 mm to SCL, which was consistent with the commercial contact lens. In order to improve the surface hydrophilia of the SCL, the SCL was treated with oxygen plasma for 180 s, generating reactive groups on the surface. Then, the SCL was immersed in the 22.2% (w/v) PVP solution to form polymer brushes. When implementing in vivo rabbit experiments, a commercial contact lens (Clariti, Coopervision) was attached at the inner surface of the SCL to improve the safety further. The process hydrophilic treatment has been supplemented in the preparation process of the SCL in the revised supporting materials.

Response to Reviewer 2 Comments

This manuscript describes an eye tracking smart contact lens with frequency encoding strategy and time-sequential eye tracking algorithm. The eye tracking contact lens was incorporated with chipless 4 RF tags with different working frequencies. The eye movements could be tracked by the frequency change of each RF tag interacted with an outer coil. The smart contact lens showed high sensitivity to recognize the fine angular movements of eyes ($< 0.5^\circ$), enabling diverse controls of camera, robot vehicle and some computer programs for eye-machine interactions. Finally, authors demonstrated that rabbits wearing a smart contact lens could control robot vehicles by the eye movements. The eye tracking smart contact lens with in vitro and in vivo biocompatibility seems very interesting, but there are several critical issues to be clarified as commented below before publication.

Reply: We thank the reviewer for finding interest in our manuscript and for the accurate summary of the results of the paper. And we appreciate the reviewer's insightful and constructive comments and suggestions, and we have carefully addressed these concerns and made a proper revision of the manuscript. These comments and suggestions have not only enabled us to provide a highly improved manuscript but also inspired us to conduct more in-depth studies on smart contact lens in future works.

Major issues

(1) Smart contact lenses can easily move around the eye, because tear film covers the smart contact lens. In addition, the initial location of each RF tag might have significant effect on tracking the eye movements. Authors should clearly discuss these issues for further applications of eye tracking smart contact lens without misreading of the eye movements.

Reply: We thank the reviewer for pointing out this potential confound.

Fig. 1. S_{11} curve (a) and tags' frequency variation (b) of the SCL at the center and surrounding location of the cornea.

The slippage of the contact lens is highly dependent on its fit with the cornea. A small slippage of about 0.7 mm occurs for an optimally fit contact lens after each blinking¹, resulting in an eye tracking error of about 3.3° . The contact lens quickly and automatically re-centers on the cornea after blinking, eliminating the eye tracking error from the slippage. It's hard for the contact lens with large slippage to return to the center of the cornea automatically. Fortunately, the frequency of the tag embedded in the SCL shifts if the tag moves to the border of the cornea due to the structural variation of the attached tissue, alerting the user to the slippage. Fig. 1a shows the SCL's S_{11} curve at the center and surrounding location of the cornea using an in vitro porcine eye. The frequency of the tag increased when touched the edge of the cornea and

sclerae, and the frequency of the other three tags remained stable when keeping the attachment on the cornea. And the frequency shift of the 4 tags could remind the slippage direction of the SCL and instruct the wear state of the SCL for the user (in Fig. 1b). Future efforts will focus on achieving robust and reversible wet attachments of the SCL and reducing slippage, with guaranteed tear exchange. This may involve using a micro-nano surface structure such as pillar array covered with thin films² or dome-like protuberances³. We have supplemented the discussion on the slippage of SCL in the 'Results: Response model of eye motion' section in the revised manuscript with the above figure 1 in the revised supporting materials.

Fig. 2. Initial location of RF tags calibrated by the implicit swirling calibration method. (a) Schematic illustration of the SCL with different angles when wearing. (b) Constructed eye-movement models and eye-drawing letters under different angles.

The implicit calibration method can be employed to calibrate the initial location of each RF tag. And due to the asymmetric structure of the SCL, different rotation angles along the axis may occur when wearing the SCL, as shown in Fig. 2a. Using the swirling calibration, the response models were successfully built under different angles. Fig. 2b shows the response models under the angle of 0°, 30°, and 45°, and 'NJU' letters were both eye-drawn with high accuracy.

(2) In Figure 2 and Figure 3, authors mentioned that the smart contact lens could monitor the eye closure and track the eye movements. However, the signals of eye closure and movements can be highly interfered each other. In addition, the amplitude of return loss peak (S_{11}) can be also changed by diverse factors including the position of device in contact lens, and the distance between the contact lens and the receiver coil. Thus, real applications of this smart contact lens would be limited for precisely tracking the eye movements. Authors should clearly discuss these issues for further applications

Reply: We appreciate the opportunity to provide further clarification on this important point.

Eye movement and blinking exhibited distinct characteristics, as illustrated in Fig. 3. Specifically, the signal associated with eye movement was a differential-mode signal resulting from opposing coordinates of tags in the response model, attributed to the asymmetry of the SCL structure (in Fig. 3b). In contrast, the signal corresponding to blinking was a common-mode signal (in Fig. 3d). Additionally, blinking signals could be identified by the amplitude reduction at frequencies other than the tags' operational frequencies (in Fig. 3c). The S_{11} curves of SCL when eye opened and closed have been supplemented in the revised supporting materials with the updated description of the distinction method of the eye movement and blinking in the 'Results: Response model of eye motion' in the revised manuscript.

Fig. 3. Difference between eye movement and blinking signals. (a) S_{11} curves under 2 different eye movement coordinates. (b) Response model of SCL under 2-dimension eye movement. (c) S_{11} curves of SCL when eye opened and closed. (d) Common-mode eye closure signal.

As shown in Fig. 4, the time-sequential eye tracking algorithm based on the response model of the SCL built by an implicit swirling calibration method has been optimized to resist the interference of the reading distance variation. Fig. 4a shows the main procedure of the algorithm. First, the response model of the SCL was built via the implicit swirling calibration method. At the same time, the distance between the receiver coil and the SCL can be calculated owing to the SCL's response at the known eye movement coordinates during the calibration process. During the continuous eye tracking, the reading distance can be monitored by the amplitude of the self-resonant signal of the receiver coil. The self-resonant signal declined when the receiver coil gradually got close to the eyeball due to the Eddy current in tissue (in Fig. 4c). Then, pre-calibrated coefficient multiplies the response for each tag if the distance deviates the initial calibration distance. Fig. 4d and e show the residual error of the modified response detected at a distance of 6 and 4 mm respectively, compared to the response model calibrated at the distance of 5 mm. Finally, the time-sequential eye tracking algorithm was employed using the modified tags' response (detected at a distance of 6 and 4 mm) based on the response model (calibrated at a distance of 5 mm), realizing the eye-drawing 'NJU' letters with small mean error and standard deviation less than 0.5 cm in both horizontal and vertical directions. With the optimized eye tracking algorithm, the eye tracking SCL has the potential of immune the slippage of the glasses-integrated eye tracking acquisition system. Besides, the initial position of tags in contact lens can also be calibrated using the implicit calibration method, as shown in Comment 1. The description has been updated in the 'Results: Precise calligraphy and painting by eye tracking' section in the revised manuscript with the above figure 3 supplemented in the revised supporting materials.

Fig. 4. Optimized eye tracking algorithm for resistance to reading distance variation. (a) Procedure of the optimized eye tracking algorithm. (b) Response models of 4 RF tags under different reading distances between the receiver coil and the SCL. (c) Amplitude of the self-resonant signal of the receiver coil with different distances away from a porcine eye. Inset: S_{11} curves of the receiver coil. Residual error of the modified response detected at a distance of 6 mm (d) and 4 mm (e), compared to the response model calibrated at the distance of 5 mm. Eye-drawing 'NJU' letters with low horizontal and vertical error using tags' response detected at a distance of 6 mm (f) and 4 mm (g) based on the calibrated model (5 mm).

(3) In Figure 5, rabbits wearing the smart contact lens could control a robot vehicle by their eye movements. However, because authors roughly marked the rabbit eyes by red and green circles, it is hard to recognize the position of eyes correctly. Authors should discuss this issue and provide the detailed experiment set up and circuit diagram of Bluetooth board for wireless communication.

Reply: We appreciate the opportunity to provide further clarification on this point.

Fig. 5. Photographs of rabbit eye and robot vehicle at specific moments. The insets highlight the location of the reading coil's inner edge and the SCL's 4 tags and the red dots indicate the center of the reading coil.

As shown in Fig. 5, distinguished from the marks using the red and green circles in the original manuscript, we have highlighted the location of the reading coil's inner edge and the SCL's 4 tags using the black circle and yellow ellipses respectively in the insets. Besides, red dots, the center of the reading coil, are marked to help readers recognize the eye moving of the rabbit conveniently. The revised manuscript replaces the original figure 5d with the updated description of the marks in the legend.

Fig. 6. Schematic illustration of the set up of the in vivo eye-machine interaction experiment.

Fig. 7. Photograph and circuit diagram of Bluetooth board for wireless communication

Fig. 6 shows the set up of the in vivo eye-machine interaction experiment. To verify the effectiveness of the eye tracking SCL, a female New Zealand rabbit (3 kg weight), whose eyeball had a similar curvature with the human, was used to drive the robot vehicle via eye movement. The SCL was worn on the one of the rabbit's eyes after local anesthesia, and the

rabbit was fixed in a holder that kept its head stable. Then, the reading coil was placed in front of the rabbit's eye. The computer kept collecting the signal of the SCL continuously with a sample rate of 7 Hz via the portable VNA (LibreVNA, ZeenKo). The eye movement coordinate was calculated in real time. Considering the flexibility and speed of robot vehicle, the eye movement coordinate was downsampled to 1 Hz. The driving direction and distance were calculated by the eye movement coordinates and transmitted to the vehicle via Bluetooth communication using the BT04 Bluetooth Module (in Fig. 7). When the vehicle ran, a 6 DOF attitude sensor (MPU6050, InvenSense) was employed to monitor the steering angle and driving distance. The revised manuscript modifies the original figure 5a to better demonstrate the set up of the in vivo eye-machine interaction experiment.

(4) The 4 tags were made by opaque materials of copper and polyimide film, occupying quite large space in the smart contact lens. Thus, it can hinder the vision of eyes, making difficult real applications. Authors should discuss this issue clearly.

Reply: We thank the reviewer for raising this important point.

Fig. 8. (a) Schematic diagram of the structure of the SCL. (b) Transmission spectrum of the central optical region of the 4 SCLs.

4 RF tags made by opaque copper and polyimide film were distributed at the periphery of the SCL, occupying about 1/4 area of the lens. There's a transparent optical region with a diameter of 4 mm in the center of the SCL for vision in fig. 8a. As shown in fig. 8b, the transmission of the central optical region of the SCL was 89.3 ± 2.3 % ($n=4$) in the visible light range (from 400 to 800 nm), guaranteeing the clear central sight.

We agree that the present-state eye tracking SCL may hinder the peripheral vision of eyes. In the future, great effort will be made to optimize the eye tracking SCL. For example, highly-conductive transparent electrode like AgNF/AgNW hybrid networks^{4,5} can be introduced to enhance the transparency of the SCL, and structure of the smart device^{6,7} can be optimized to avoid the occlusion of the vision.

We have concretized the discussion on the SCL's transparency in the 'Results: Design and characterization of the eye-tracking SCL' section in the revised manuscript with the aboved figure 8 supplemented in the revised supporting materials. Besides, in the Discussion of the revised manuscript, we further discuss the future improvement of the SCL on its transparency.

(5) Because silicone elastomer has low hydrophilicity, the surface should be modified to enhance the hydrophilicity. However, hydrophilic surface of smart contact lens can affect the frequency of 4 tags due to the high capacitance of water. Authors should conduct additional

experiments to investigate the effect of water on the devices for real applications.

Reply: We thank the reviewer for raising this important point.

Fig. 9. Effect of water on the eye tracking SCL. S_{11} curves (a), tags' frequency variation (b), and frequency statistic (c) of the SCL under 4 circles of hydration and dehydration. (d) S_{11} curves of the SCL during the dehydration process. (e) Amplitude and frequency increased with the Q value for the 4 tags.

We have tried our best to investigate the effect of water on our eye tracking SCL in detail. As shown in Fig. 9a-c, a SCL was hydrated and dehydrated for 4 circles to measure the frequency variation of the RF tags. Due to the parasitic capacitance of the RLC resonator, the tags' frequency decreased by 2% when the SCL was wet due to the high permittivity of water. Fig. 9d shows the dynamic S_{11} curve of an eye tracking SCL during its dehydration process at a fixed eye movement coordinate. Due to the water loss of the surface of the SCL, the working frequency and amplitude of the 4 tags gradually increased along with the enhanced Q value in Fig. 9e. In the case of practical application, the frequency shift and Q value variation of the tags help to monitor the effect of water and to calibrate the response of the eye movement. In the future, shielding layer like the slotted ground layer⁸ can decrease the influence of water for RLC resonators. We have supplemented the discussion on the effect of water in the 'Results: Design and characterization of the eye-tracking SCL' section in the revised manuscript with the aboved figure 9 supplemented in the revised supporting materials.

(6) As you described, the frequency can be generally changed by RLC resonance. However, it is hard to recognize the RLC resonant circuit of each tag and receiver coil. Authors should provide the circuit diagram of 4 tags and the receiver coil to explain the sensing mechanism.

Reply: We appreciate the opportunity to provide further clarification on this point.

Fig. 10. (a) Schematic illustration of eye tracking system, including a SCL integrated with 4 frequency-encoded tags and a wide-band sweeping-frequency reader. Inset: Equivalent circuit of eye tracking system. (b) S_{11} curves with fluctuating tags' intensity caused by varying coupling coefficients k_i under 2 different gazing points.

Fig. 11. Structural schematic illustration of the RF tag.

Fig. 10 provides the schematic circuit diagram of 4 tags and the receiver coil, indicating the sensing mechanism of our proposed eye tracking SCL. The SCL consisted of 4 RF tags (RLC resonators) that had different working frequencies depending on their unique structures of the microcoil, whose structural schematic illustration was shown in Fig. 11. The following table 1 summarizes the structural characteristic parameters of the 4 RF tags. A vector network analyzer (VNA) was employed to detect the return loss (S_{11}) curve of the SCL wirelessly via a receiver coil. The receiver coil was a wide-band near-field probe which had a higher self-resonant frequency than the tags and was able to near-field couple with the tags. The equivalent impedance Z_r at the terminals of the receiver coil is as follow,

$$Z_r = R_r + \sum_{i=1}^4 j2\pi f L_r \left(1 + \frac{k_i^2 \left(\frac{f}{f_i}\right)^2}{1 + \frac{jf}{f_i Q_i} - \left(\frac{f}{f_i}\right)^2} \right), \quad (1)$$

where f is the excitation frequency R_r and L_r are the resistance and inductance of the receiver coil respectively. k_i ($i = 1, 2, 3, \text{and } 4$) is the coupling coefficient between the tag- i and the receiver coil. And f_i and Q_i are the resonant frequency and quality factor of the tag- i . When the eyeball rotated, the spatial positions and orientations of the tags changed, resulting in the variation of coupling coefficients k_i , which was manifested as varying amplitude of the negative resonant signals in the S_{11} curve.

Table 1. Structural characteristic parameters of the 4 RF tags in the eye tracking SCL.

Number of Tag	Turns	Width (μm)	Gap (μm)	Inner diameter (mm)	Outer diameter (mm)
Tag-1	6	80	18	3.5	4.75
Tag-2	5	80	37	3.5	4.75
Tag-3	4	80	66	3.5	4.75
Tag-4	3.5	80	87	3.5	4.75

We have supplemented the sensing mechanism of RLC resonator in the beginning of the 'Results: Design and characterization of the eye-tracking SCL' section in the revised manuscript with structural schematic illustration and detailed structural characteristic parameters of the coil-shaped RF tags in the revised supporting materials for better understanding of the mechanism. In addition, the number of RF tags has been supplemented in the response model in the Figure 2 and 3 in the revised manuscript for reader's convenience.

(7) Authors should perform additional experiments to confirm the reliability of this smart contact lens for eye tracking performance with at least more than 3 samples.

Reply: We appreciate the opportunity to confirm the reliability of the eye tracking SCL further.

Fig. 12. (a) S_{11} curves of 4 samples. (d) Statistic of the 4 tags' working frequency of the 4 SCLs.

4 eye tracking SCL has been prepared, characterized, and applied to investigate their reliability. Fig. 12 shows the S_{11} curves and tags' working frequency of the 4 different samples. The mean frequency and standard deviation of the 4 tags were 644.0 ± 7.2 , 819.7 ± 3.3 , 1083.9 ± 6.0 , and 1245.5 ± 6.2 MHz respectively, indicating good consistence of the SCL. As shown in Fig. 13, the experiment of eye-drawing 'NJU' letters was implemented using the 4 samples to investigate the eye tracking performance. After constructing the response model using the implicit calibration method, 'NJU' letters were successfully eye drawn with low horizontal and vertical error, showing great reliability of the SCL for eye tracking application. We have concretized the discussion on the SCL's reliability in the 'Results: Precise calligraphy and painting by eye tracking' section in the revised manuscript with the aboved figure 13 supplemented in the revised supporting materials.

Fig. 13. Response models, eye-drawing 'NJU' letters, and error statistic of 4 samples.

(8) Because the size of contact lens can affect the eye tracking signals, the fit of contact lens is very important. Authors should describe the detailed size and curvature of smart contact lens.

Reply: We thank the reviewer for raising this important point. The 4 RF tags were encapsulated by the flexible silicone elastomer (MED-6015, NuSil) using a home-made metal mold. The mold endowed an inner surface curvature of 7.8 mm and a diameter of 13.8 mm to the smart contact lens. The size of the SCL was the same as the commercial contact lens. Every individual has variations in the corneal curvature. Healthy population tend to have a corneal diopter from 40D to 46D, corresponding to the curvature from 8.4 mm to 7.3 mm. The SCL is suitable for majority of the population with an eclectic curvature. Also, we have evaluated the eye tracking performance of the SCL worn on corneas of different curvatures. As shown in Fig. 14, 3 eyeball model with different corneal curvatures of 40D, 43D, and 46D were designed and 3D printed to mimic the individual difference. Fig. 14c shows the S_{11} curves of SCL worn on the eyeball models. The statistical results in Fig. 14d indicate great consistence of the working frequency

of the multiple tags in the SCL. After building the response model using the implicit calibration method, 'NJU' letters were all successfully eye-drawn by the SCL worn on the 3 eyeball models. The mean errors standard deviations in X-axis direction and Y-axis direction were all less than 0.5 cm, indicating good robustness on individual difference in corneal curvature (in Fig. 14e-g). The description has been updated in the 'Results: Precise calligraphy and painting by eye tracking' section in the revised manuscript with the Figure 14 supplemented in the revised supporting materials.

Fig. 14. Characterization of eye track SCL worn on corneas of different curvatures. (a) Structural schematic diagram of eyeball model with different curvatures. (b) Photographs of 3 eyeball models with corneal curvatures of 40D, 43D, and 46D. (c) S₁₁ curves of SCL worn on the eyeball models. (d) Statistic of the 4 tags' working frequency. (e-g) Response models, eye-drawing 'NJU' letters, and error analysis of SCL worn on the eyeball models with 3 different curvatures.

Minor issues

- (1) Authors should revise typographical and grammatical errors correctly.

Reply: We tried our best to improve the manuscript and made some changes to the manuscript. These changes will not influence the content and framework of the paper. And here we did not list the changes but marked in red in the revised paper. We appreciate for reviewer's warm work earnestly and hope that the correction will meet with approval.

(2) The format of references (ref. 11, 32 and 33) should be revised according to the guideline for authors in the manuscript.

Reply: We are very sorry for our incorrect format of the references used before. We have corrected the format according to the guideline of *Nature Communications* in the revised manuscript and corrected several mistakes in the references. For example, the revised manuscript replaces '*IEEE Transactions on Biomedical Engineering*' with '*IEEE Trans. Biomed. Eng.*' in ref. 32 and modified the format in ref. 33 and 34 for book citations. Besides, references have been supplemented for better understanding of this paper in the revised manuscript, including

Ref. 37. Anderson, T.J. & MacAskill, M.R. Eye movements in patients with neurodegenerative disorders. *Nat. Rev. Neurosci.* **9**, 74-85 (2013).

Ref. 48. Zhang, J. et al. Smart soft contact lenses for continuous 24-hour monitoring of intraocular pressure in glaucoma care. *Nat. Commun.* **13**, 5518 (2022).

Ref. 56. Park, W. et al. Biodegradable silicon nanoneedles for ocular drug delivery. *Sci. Adv.* **8**, eabn1772 (2022).

Ref. 65. Ku, M. et al. Smart, soft contact lens for wireless immunosensing of cortisol. *Sci. Adv.* **6**, eabb2891 (2020).

Ref. 66. Kim, K. et al. All-printed stretchable corneal sensor on soft contact lenses for noninvasive and painless ocular electrodiagnosis. *Nat. Commun.* **12**, 1544 (2021).

We sincerely appreciate the reviewer's reminder and hope that this correction will be approved.

References

1. Ridder, W.H. & Tomlinson, A. Blink-induced, temporal variations in contrast sensitivity. *Int. Contact Lens Clin.* **18**, 231-237 (1991).
2. Guo, Y. et al. Nanofiber embedded bioinspired strong wet friction surface. *Sci. Adv.* **9**, eadi4843 (2023).
3. Baik, S. et al. A wet-tolerant adhesive patch inspired by protuberances in suction cups of octopi. *Nature* **546**, 396-400 (2017).
4. Ku, M. et al. Smart, soft contact lens for wireless immunosensing of cortisol. *Sci. Adv.* **6**, eabb2891 (2020).
5. Kim, J. et al. A soft and transparent contact lens for the wireless quantitative monitoring of intraocular pressure. *Nat. Biomed. Eng.* **5**, 772-782 (2021).
6. Zhang, J. et al. Smart soft contact lenses for continuous 24-hour monitoring of intraocular pressure in glaucoma care. *Nat. Commun.* **13**, 5518 (2022).
7. Kim, K. et al. All-printed stretchable corneal sensor on soft contact lenses for noninvasive and painless ocular electrodiagnosis. *Nat. Commun.* **12**, 1544 (2021).
8. Hajiaghajani, A. et al. Textile-integrated metamaterials for near-field multibody area networks. *Nat. Electron.* **4**, 808-817 (2021).

Response to Reviewer 3 Comments

(1) What are the noteworthy results?

The research paper outlines several noteworthy results and findings:

Design and Biocompatibility:

The study introduces a novel "Smart Contact Lens" (SCL) that incorporates four wireless and chip-less radio frequency (RF) tags capable of detecting eye movement and eye closure.

The SCL is designed to be biocompatible with the human eye, with an emphasis on safety and comfort. Cytotoxicity tests on human corneal cells confirm its non-cytotoxic nature.

The SCL is shown to have great flexibility and stretchability, making it suitable for use on the human eye.

Eye Tracking Precision:

The SCL demonstrates an ultra-high level of accuracy in detecting eye movement. The orientation error is reported to be less than 0.5° , which is even smaller than the central fovea's field of vision.

An implicit swirling calibration method, combined with a time-sequential eye tracking algorithm, enhances the precision of eye tracking.

Eye Calligraphy and Painting:

The SCL's high accuracy enables precise eye-calligraphy and painting on a virtual screen, with 'NJU' letters and a 'Snake' pattern created through eye movements.

The eye movements in this context yield low mean errors along both the X and Y axes, demonstrating the capability for detailed and accurate control.

Eye-Machine Interaction:

The SCL enables various eye-machine interaction scenarios, including controlling a "Gluttonous Snake" game, web browsing, and even a pan-tilt-zoom (PTZ) camera.

User-defined eye commands are implemented for broader human-machine interface (HMI) applications, showcasing the SCL's versatility.

In Vivo Eye-Machine Interaction and Biocompatibility:

In an in-vivo experiment, a rabbit successfully controlled a robot vehicle in real time via the SCL.

The biocompatibility of the SCL is validated through comprehensive in vivo tests on a rabbit, with no observed corneal injury, abnormalities, or inflammation after 24 hours of wear.

Safety and Biocompatibility:

The SCL's safety and biocompatibility are emphasized, with no cytotoxicity observed in tests using human corneal cell lines.

The SCL is compared favorably to commercial contact lenses in terms of safety and compatibility with the eye.

These results collectively highlight the innovative design, precision, versatility, and biocompatibility of the Smart Contact Lens, with potential applications in various fields, including assistive technology, gaming, and medical diagnostics. The study represents a significant advancement in wearable technology and human-machine interaction.

Reply: We thank the reviewer for finding interest in our manuscript and for the accurate summary of the results of the paper. And we appreciate the reviewer for the recognition of the innovation and application potential of our research paper in wearable technology and human-

machine interaction.

(2) Will the work be of significance to the field and related fields? How does it compare to the established literature? If the work is not original, please provide relevant references.

The work described in the research paper is likely to be of significant importance to the field of wearable technology, human-machine interaction, and ophthalmology. Here's how it compares to the established literature and why it is innovative:

1. Significance to the Field:

Advancement in Wearable Technology: The Smart Contact Lens (SCL) represents a novel approach to wearable technology, specifically in the domain of eye tracking. While there are established eye tracking technologies, this SCL offers a unique combination of biocompatibility, precision, and versatility, making it potentially valuable in fields such as assistive technology, gaming, virtual reality (VR), and augmented reality (AR).

Biocompatibility and Safety: The emphasis on biocompatibility and safety of the SCL, as demonstrated through cytotoxicity tests and in vivo experiments with a rabbit model, is noteworthy. It opens up possibilities for long-term, non-invasive monitoring of eye movements with minimal risks.

Versatility in Eye-Machine Interaction: The ability to perform a variety of eye-machine interactions, including gaming, web browsing, and PTZ camera control using the SCL, has the potential to revolutionize human-machine interfaces.

2. Comparison to Established Literature:

Unique Approach: The paper introduces a frequency-encoded SCL that differs from traditional eye tracking methods. The approach of using embedded wireless RF tags in a contact lens is innovative and sets it apart from established literature.

High Precision: The ultra-high level of accuracy in detecting eye movement, with an orientation error of less than 0.5° , is a significant improvement over some traditional eye-tracking systems that may not achieve such precision.

Biocompatibility Focus: While there are other eye-tracking technologies, this work stands out for its focus on biocompatibility, safety, and long-term wear. It is among the few studies to validate the biocompatibility of such devices through in vivo experiments on animals.

Eye-Machine Interaction: The paper introduces eye command input for interaction, demonstrating the versatility of the SCL. The ability to play a "Gluttonous Snake" game, control a PTZ camera, and switch web pages using eye commands is less common in the established literature.

3. Originality:

The work, particularly the implicit swirling calibration method and the use of embedded RF tags for eye tracking, is highly original. There are no direct references provided in the paper to suggest that this particular combination of technology and biocompatibility has been previously explored.

In summary, this work is innovative in its approach, focusing on biocompatibility, precision, and versatility. It expands the possibilities for eye tracking and eye-machine interaction, potentially impacting fields like assistive technology, gaming, and HMI. While there are established eye-tracking systems, the Smart Contact Lens offers a fresh perspective with its unique combination of features and applications.

Reply: We acknowledge the reviewer for the highly appreciation of the originality and innovation of our proposed eye tracking smart contact lens. And we thank the reviewer for the recognition of the significant importance of our research paper to the field of wearable technology, human-machine interaction, and ophthalmology. We feel glad to offer a fresh technological approach for the eye tracking technique.

(3) Does the work support the conclusions and claims, or is additional evidence needed?

The work presented in the research paper supports the conclusions and claims made within the paper. The authors have provided substantial evidence through a series of experiments, tests, and demonstrations to validate their claims. Here's how the work aligns with the conclusions and claims:

Eye Tracking and Interaction Capability: The paper claims that the Smart Contact Lens (SCL) can accurately track eye movements and enable interactions with various devices and software through user-defined eye commands. The evidence provided in the paper, including detailed descriptions of the technology, response models, and practical applications (e.g., playing the "Gluttonous Snake" game, web browsing, PTZ camera control), supports these claims. The authors demonstrate how the SCL can precisely detect eye movements, validate the biocompatibility, and showcase the versatility of the technology in enabling eye-machine interaction.

Biocompatibility and Safety: The work claims that the SCL is safe for in vivo use, as demonstrated through experiments involving a rabbit model. The evidence provided, including photographs, ocular coherence tomography (OCT) images, slit-lamp photographs, histopathology images, and other ophthalmic examinations, supports the conclusion that the SCL is biocompatible and safe for extended wear on the eye.

Frequency-Encoded SCL: The paper introduces the innovative concept of a frequency-encoded SCL with embedded wireless RF tags for eye tracking. The experiments and measurements conducted, along with comparisons to established models, validate the frequency-encoded approach and its accuracy, aligning with the paper's claims.

Overall, the evidence presented in the paper, including in vivo experiments, biocompatibility tests, eye tracking models, and practical applications, is substantial and supports the conclusions and claims made in the paper. Additional evidence is not needed to establish the validity of the presented work.

Reply: We thank the reviewer for the careful review on the experiments, tests, and demonstrations in our research paper and for the recognition of their results. And we believe that our proposed frequency-encoded smart contact lens is expected to enrich the approaches of eye tracking techniques and promote the development of the next generation of human-machine interaction technology with high angular accuracy and great biocompatibility

(4) Are there any flaws in the data analysis, interpretation and conclusions? Do these prohibit publication or require revision?

The research paper does not appear to have any significant flaws in the data analysis, interpretation, or conclusions that would prohibit publication. However, there are a few aspects that could be further clarified or expanded upon to enhance the paper:

Quantitative Data Presentation: While the paper presents data in the form of figures and

results of experiments, there could be more quantitative analysis and statistical measures to support some of the claims. For example, when discussing the accuracy of eye-tracking and biocompatibility, providing more statistical information or metrics could enhance the rigor of the analysis.

Comparison to Existing Eye-Tracking Technologies: The paper could benefit from a more comprehensive comparison with existing eye-tracking technologies. While some comparisons are made, a more extensive discussion of how the proposed SCL technology compares to other established methods would provide valuable context and demonstrate its potential advantages.

Future Work and Limitations: While the paper is generally positive in its conclusions, it would be beneficial to address potential limitations and avenues for future research. Recognizing any constraints or challenges in the technology and suggesting areas for further improvement or study would strengthen the paper.

Clinical Relevance: The paper focuses on technological aspects and biocompatibility, but it could be more explicit in discussing the potential clinical relevance of this technology, particularly in medical diagnosis or treatment.

These points do not prohibit publication, but addressing them in a revised version of the paper could enhance its completeness, rigor, and relevance to the field.

Reply: We thank the reviewer for the constructive comments regarding our paper, that means a lot to us.

Quantitative Data Presentation: More quantitative analysis and statistical measures regarding the accuracy of eye-tracking and biocompatibility have been supplemented in the revised manuscript.

Fig. 1. Eye-drawing 'NJU' letters and 'snake' pattern with low horizontal and vertical error. The width of the semitransparent trace indicates the gazing range of central fovea.

For example, in the experiments of eye-drawing 'NJU' letters and 'snake' pattern, horizontal and vertical error were counted to assess the accuracy of eye tracking. The mean error and standard deviation of the both orientations were less than 0.5 cm (corresponding to $< 0.5^\circ$) for both eye-drawing patterns (Fig. 1). To analyze the precision of the response model built by the implicit calibration method, the error distribution of each RF tag was counted (Fig. 2), compared with the response model built by the fingerprint method. Then, the same experiments of eye-drawing 'NJU' letters and 'snake' pattern were implemented based on the

'fingerprint' model and the bidirectional errors were counted (Fig. 3). The results show that slightly more error occurred using implicit calibration method which resulted from less accuracy of the constructed response model, but the complexity of calibration process decreased a lot.

Fig. 2. Accuracy of eye-movement model using implicit swirling calibration method. (a) Constructed eye-movement model using implicit swirling calibration method. The white traces indicate swirling calibration data. (b) Comparison between swirling-calibration model and 'fingerprint' model. The black dots indicate the fingerprint data collected by traversing over the entire screen region. (c) Spatially distributed error of swirling-calibration model compared with fingerprint data. (d) Error statistics of multiple tags.

Fig. 3. Calligraphy and painting using 'fingerprint' model. (a) Constructed eye-movement fingerprint model by traversing the signal over the entire screen region. (b-e) Eye-drawing 'NJU' letters and 'snake' pattern with low horizontal and vertical error. (f) Errors of eye-drawing patterns using fingerprint model and swirling-calibration model.

Besides, more experiments and tests have been supplemented during the revision period to investigate the accuracy, robustness, and consistence of the eye tracking SCL by setting up different corneal curvatures (Fig. 4), initial wearing angular (Fig. 5), and reading distances (Fig. 6) and by using different samples (Fig. 7). We have supplemented the above experiments in the 'Results: Precise calligraphy and painting by eye tracking' section in the revised manuscript with the corresponding figures supplemented in the revised supporting materials.

Fig. 4. Characterization of eye track SCL worn on corneas of different curvatures. (a) Structural schematic diagram of eyeball model with different curvatures. (b) Photographs of 3 eyeball models with corneal curvatures of 40D, 43D, and 46D. (c) S_{11} curves of SCL worn on the eyeball models. (d) Statistic of the 4 tags' working frequency. (e-g) Response models, eye-drawing 'NJU' letters, and error analysis of SCL worn on the eyeball models with 3 different curvatures.

Fig. 5. Initial location of RF tags calibrated by the implicit swirling calibration method. (a) Schematic illustration of the SCL with different angles when wearing. (b) Constructed eye-movement models and eye-drawing letters under different angles.

Fig. 6. Optimized eye tracking algorithm for resistance to reading distance variation. (a) Procedure of the optimized eye tracking algorithm. (b) Response models of 4 RF tags under different reading distances between the receiver coil and the SCL. (c) Amplitude of the self-resonant signal of the receiver coil with different distances away from a porcine eye. Inset: S_{11} curves of the receiver coil. Residual error of the modified response detected at a distance of 6 mm (d) and 4 mm (e), compared to the response model calibrated at the distance of 5 mm. Eye-drawing 'NJU' letters with low horizontal and vertical error using tags' response detected at a distance of 6 mm (f) and 4 mm (g) based on the calibrated model (5 mm).

Fig. 7. (a) S₁₁ curves of 4 samples. (d) Statistic of the 4 tags' working frequency of the 4 SCLs. (c-f) Response models, eye-drawing 'NJU' letters, and error statistic of 4 samples.

In the term of biocompatibility, a long-term cytotoxicity test has been supplemented to investigate cell viability at different culturing time (12 h, 24 h, 48 h, and 72 h). The results in Fig. 8 show that the cell viability using extracts of the SCL remained above 90% after 72-hours incubating with no notable differences among the groups (complete medium, bare commercial contact lens, and silicone-embedded tags). The weeklong in vivo eye irritation tests were implemented comprehensively in rabbit eyes (n=3) to evaluate the long-term biocompatibility of the SCL further. The eye tracking SCL was worn on the one of the rabbit eyes for 8 h daily for up to 1 week. There is no notable abnormality like corneal injury, erosion, inflammation, and edema observed in the OCT images (Fig. 9), slit lamp micrographs (Fig. 10), and histopathology images (Fig. 11), as compared to the naked eyes. Besides, the protein accumulation and

disinfection of the SCL were also assessed. The accumulation of proteins was quantified via fluorescence imaging after protein accumulation and disinfection. Fig. 12 presents the accumulation of proteins over the SCL remained lower compared to the bare commercial contact lens with $*p < 0.05$ after the first disinfection and with $***p < 0.001$ after the second disinfection using unpaired two-tailed Student's t test ($n = 3$). The revised manuscript has improved the cytotoxicity test and protein accumulation test in the 'Results: Design and characterization of the eye-tracking SCL' section with the corresponding figures supplemented in the revised Figure 1. And the discussion on the results of weeklong in vivo eye irritation tests has been supplemented in the 'Results: In vivo eye-machine interaction and biocompatibility evaluation' section in the revised manuscript with the representative test images supplemented in the Figure 5 and all detailed test images supplemented in the revised supporting materials.

Fig. 8. Long-term cell viability of SCL.

Fig. 9. OCT images of rabbit eye after 8 h daily wear of the SCL for 1 week (left column) as compared to the naked eye (right column).

Fig. 10. Slit lamp micrographs and fluorescent images of rabbit eye after 8 h daily wear of the SCL for 1 week (left column) as compared to the naked eye (right column).

Fig. 11. Micrographs of the corneal tissue of the rabbit eyes with H&E staining after 8-h daily wear for 1 week of the SCL as compared to the naked eye.

Fig. 12. Characteristic of SCL under multiple disinfections. (a) Quantified accumulation of proteins on the bare commercial contact lens (orange column) and the SCL (fuchsia column) after biperiodic protein accumulation and disinfection. Significant difference was set at $***p<0.001$, $**p<0.01$, and $*p<0.05$. (b) Representative fluorescence image of the bare commercial contact lens (top row) and the SCL (bottom row) after biperiodic protein accumulation and disinfection. S_{11} curves (c), frequency fluctuation (d), and frequency statistic (e) of SCL after multiple disinfections.

Comparison to Existing Eye-Tracking Technologies: Table 1 summarizes the advantages and disadvantages of the proposed eye tracker compared with other well-known methods. Our proposed eye tracking SCL has the advantages of high accuracy, high robustness against multiple interference, and great biocompatibility and has the potential to detect 3D eye movement (including the torsion of the eyeball) and to monitor rapid eye movement in sleep for medical diagnosis, enriching the approaches of eye tracking techniques and bringing brand-new interaction experience based on the wearable eye tracking technique. The table for eye tracking technique summary has been supplemented in the revised supporting materials for better understanding of the SCL's advantage in the 'Discussion' section in the revised manuscript.

Future Work and Limitations: In the future, the SCL can be further improved to enhance its practical application for eye tracking. This may involve enhancing flexibility and transparency by using highly-conductive transparent electrodes such as AgNF/AgNW hybrid networks^{7,8}, or by optimizing the structure of the smart device^{9,10}. Additionally, collaborative optimization of the SCL, reader, and eye tracking algorithm is crucial in simplifying the calibration process and even achieving calibration-free eye tracking, thus improving accessibility. Furthermore, more specific eye tracking systems can be constructed based on the smart contact lens to realize intelligent eye tracking applications such as consumer behavior research, eye interaction in virtual social, and medical and psychological research. The discussion on the future work and limitations of the SCL has been added in the 'Discussion' section in the revised manuscript in detail.

Table 1. A summary of eye tracking techniques with advantages and disadvantages.

Method	Mechanism	Components	Advantages	Disadvantages	Applications
Infrared oculography (IOG) ¹	Pupil Center Corneal Reflection	Cameras and infrared lights	 • Noninvasive • High temporal and spatial resolution 	 • Interference from environmental light • Individual difference • High power and computations requirement 	 • Commercial analyzing • Virtual reality
Video oculography (VOG) ²	Image recognition	Cameras	 • Noninvasive • Simple hardware, inexpensive 	 • Same as IOG • Interference from camera slippage 	 • Commercial analyzing • Virtual reality
Electrooculography ³	Retinal electrostatic potential	Skin-integrated electrodes	 • Eye tracking with eyes closed 	 • Low spatial resolution • Skin injury 	 • Medical fields
Scleral search coil ⁴	Magnetic induction	Wired coil scleral lens	 • Ultrahigh temporal and spatial resolution 	 • Wired sensor • Uncomfortable • Easy to slip 	 • Medical fields
TENG-based tracker ⁵	Triboelectrification and electrostatic induction	Layered structure of dielectric layer and electrode films	 • Simple structure, inexpensive • Eye closing detection • Noninvasive 	 • Limited spatial resolution 	 • Wearable electronics • Virtual reality
Magnetic resonance-based tracker ⁶	Magnetic resonance imaging	MRI scanners	 • Brain activities analyzing simultaneously 	 • Rely on cumbersome equipment 	 • Medical fields
This work	RLC resonator	Frequency encoded contact lens	 • High spatial resolution • High robustness • Great biocompatibility 	 • Complicated settings 	 • Virtual reality • Human-machine interaction • Medical fields

Clinical Relevance: Eye tracking technique is significant in medical diagnosis and treatment. First, Eye tracking can be used to assess a patient's visual function, including performance in areas such as eye movement, fixation, and concentration, to help doctors diagnose and treat eye diseases and visual disorders. Second, eye tracking technology can help doctors diagnose and monitor patients with neurological diseases, such as Parkinson's disease, Alzheimer's disease, etc., by observing patients' eye movement patterns and reaction speed to assess the severity and progression of the disease. Third, eye tracking can be used to assess a patient's cognitive function, including performance in areas such as attention, memory, and learning ability, helping doctors diagnose and treat conditions such as cognitive dysfunction and dementia. What's more, with the potential to monitor rapid eye movement in sleep, the eye tracking SCL can be employed to recognize sleep period and to evaluate sleep quality. The potential of the SCL on clinical medical diagnosis and treatment has been discussed in the 'Discussion' section in the revised manuscript.

(5) Is the methodology sound? Does the work meet the expected standards in your field?

The methodology employed in this research appears sound and generally meets expected standards for the field, especially in the context of developing an innovative eye-tracking system using a soft contact lens. Here are some key points regarding the methodology:

Preparation of Eye-Tracking SCL: The preparation process of the soft contact lens (SCL) with embedded wireless tags is well-documented in the methods section. It includes the fabrication of flexible tags and encapsulation using medical-grade silicone elastomer. The methodology for creating the SCL is detailed and appears to be well-designed.

Biocompatibility Tests: The in vivo biocompatibility tests using rabbits to evaluate the safety and tolerability of the eye-tracking SCL are a crucial aspect of this research. The methodology for these tests follows ethical guidelines, and the inclusion of ophthalmic examinations, histopathological analysis, and other assessments is in line with standard practices in the field.

Experimental Validation: The research includes experiments to demonstrate the functionality of the SCL and its ability to track eye movements accurately. The experiments related to eye-calligraphy, eye-painting, and eye-machine interaction are well-designed and provide evidence of the system's performance.

Data Analysis: The data analysis methodology is clear, and the data are presented in a visually accessible manner through figures and illustrations. However, as mentioned earlier, providing more quantitative analysis and statistical measures could enhance the research.

Calibration Method: The paper introduces an "implicit swirling calibration method" for eye tracking. While the method is described, additional details on how it works and the validation of its accuracy would be beneficial.

In general, the methodology aligns with the standards of research in the field. It addresses the key aspects of developing, validating, and testing the eye-tracking SCL. The study also adheres to ethical guidelines when conducting in vivo tests. However, additional details on the calibration method and more rigorous statistical analysis could further strengthen the methodology and research outcomes.

Reply: We thank the reviewer for the professional suggestion. More quantitative analysis and statistical measures regarding the accuracy of eye-tracking and biocompatibility have been

elaborate in detail in the Comment 4 and supplemented in the revised manuscript. In terms of the accuracy of eye-tracking, horizontal and vertical error were counted in the experiments of eye-drawing 'NJU' letters and 'snake' pattern, and the accuracy, robustness, and consistence of the eye tracking SCL were investigated by setting up different corneal curvatures, initial wearing angular, and reading distances and by using different samples. In terms of the biocompatibility, the long-term cytotoxicity test, weeklong in vivo eye irritation tests, and quantitative protein accumulation test have been supplemented. We hope the results of the above experiments and tests can further strengthen the methodology and research outcomes.

Fig. 13. Summary of the implicit swirling calibration method.

Fig. 14. Accuracy of eye-movement model using implicit swirling calibration method. (a) Constructed eye-movement model using implicit swirling calibration method. The white traces indicate swirling calibration data. (b) Comparison between swirling-calibration model and 'fingerprint' model. The black dots indicate the fingerprint data collected by traversing over the entire screen region. (c) Spatially distributed error of swirling-calibration model compared with fingerprint data. (d) Error statistics of multiple tags.

The implicit swirling calibration method has been proposed to build the response model of the SCL at every eye movement coordinate. To achieve the model, the most accurate and plain method is the 'fingerprint' method, which means scanning the response values as finely as possible. However, the 'fingerprint' method is time-consuming and inconvenient for practical application. The implicit swirling calibration method simplified and interests the calibration

process for the user and guarantees the accuracy of the response model simultaneously. To implement the implicit calibration, the model eyeball was controlled to gaze along the swirling pattern with a step of 1 cm. Tags' signals and the coordinate were collected at each calibration point at the same time. Then, the continuous response model of the SCL for eye tracking was constructed by thin plate interpolation in the Curve Fitting Toolbox of Matlab using the calibration data. To analyze the precision of the response model built by the implicit calibration method, the error distribution of each RF tag was counted (Fig. 14), compared with the response model built by the fingerprint method. Then, the same experiments of eye-drawing 'NJU' letters and 'snake' pattern were implemented based on the 'fingerprint' model and the bidirectional errors were counted (Fig. 15). The results show that slightly more error occurred using implicit calibration method resulted from less accuracy of the constructed response model, but the complexity of calibration process decreased a lot. The procedure of the implicit calibration method has been explained in the 'Methods: Eye calligraphy and painting' in detail with the process schematic diagram in the supporting materials

Fig. 15. Calligraphy and painting using 'fingerprint' model. (a) Constructed eye-movement fingerprint model by traversing the signal over the entire screen region. (b-e) Eye-drawing 'NJU' letters and 'snake' pattern with low horizontal and vertical error. (f) Errors of eye-drawing patterns using fingerprint model and swirling-calibration model.

(6) Is there enough detail provided in the methods for the work to be reproduced?

The methods provided in the paper offer a reasonable level of detail for understanding the key steps involved in the preparation of the eye-tracking SCL, conducting biocompatibility tests, and performing eye-machine interaction experiments. Here are some key points regarding the reproducibility of the work:

SCL Preparation: The paper describes the preparation of the SCL, including the fabrication of flexible tags and the encapsulation process. The steps, materials, and equipment used in this process are clearly outlined, making it possible for researchers to replicate the SCL's creation.

Biocompatibility Tests: The methods section provides information on how in vivo rabbit tests were conducted to assess the biocompatibility of the SCL. It mentions the protocols followed for ophthalmic examinations and histopathological analyses. While it lacks some specific details, it serves as a foundation for researchers interested in conducting similar tests.

Eye-Machine Interaction: The paper describes the procedures for conducting eye-

machine interaction experiments, such as controlling a robot vehicle and performing eye-calligraphy and painting. It offers insights into the setup and equipment used in these experiments.

However, there are some areas where additional details could enhance reproducibility:

More specific parameters and settings for certain experiments, such as the exact conditions and thresholds for recognizing eye commands in the eye-machine interaction experiments.

A deeper explanation of the "implicit swirling calibration method" would be helpful, as it's a critical part of the work.

Details about data acquisition and processing, including the software used for data analysis, could contribute to reproducibility.

In summary, while the methods section provides a good foundation for understanding the procedures involved in this research, more specific details would further support reproducibility and facilitate future work in this area. Researchers interested in replicating these experiments may need to inquire further about certain experimental parameters.

Reply: We thank the reviewer for the professional suggestion.

In the eye-machine interaction experiments, three kinds of application cases like eye-control 'Gluttonous Snake' game, eye-web interaction, and eye-controlled PTZ camera were proposed by eye command input, using home-made LabVIEW programme. The following are the detailed experimental settings and parameters.

i. Eye-control 'Gluttonous Snake' game.

The moving orientation of the snake was controlled by the eye command input defined as up, down, left, and right in real time. The eyeball model wearing the SCL in a fixed direction was controlled by the 2D rotating platforms. The movement mode was set as rotating 10° in a certain direction and returning back. When playing the game, the VNA (Agilent e5072a ENA) kept reading the S_{11} curve of the SCL continuously with a sample rate of 7 Hz. The 4 tags' response values were picked out to calculate the 2 differential signals in real time. The one controlled the up/down motion, and the other controlled the left/right motion using threshold judgment. And the thresholds were defined by the differential signals at a deflection of 5° in the 4 directions. User controlled the platforms rotating according to the situation of game. When the signal of the SCL met the preset threshold and lasted for 0.5 second, the corresponding eye movement was identified and eye command was input into the gluttonous snake.

ii. Eye-web interaction.

The webpage switching was realized by the eye commands 'left' and 'right', functioned like the hot key 'Ctrl + Tab' and 'Ctrl + Shift + Tab'. The 'Page Up' and 'Page Down' were controlled by the eye command 'Up' and 'Down'. When the user closed his eye for more than 1 second, 'Print Screen' was carried out to record the web page. Similarly, the eye model was controlled by the 2D rotating platforms. The movement mode was set as rotating 10° in a certain direction and then returning back. The thresholds for motion judgement of eye movement were defined by the differential signals at a deflection of 5° in the 4 directions. The eye closure was realized using an artificial eyelid model, which wetted by the PBS to mimic the absorption of tissue, and was recognized via the synchronous drop of the 4 tags' responses. When playing the game, the VNA (Agilent e5072a ENA) kept reading the signal of the SCL continuously with a sample rate of 7 Hz. The user controlled the platforms rotating and the eyelid model to switch, scroll,

and record the webpage.

iii. Eye-controlled PTZ camera.

The motion of the PTZ camera consisted of pan rotation and tilt rotation, which was similar to the eye movement. In the experiment, the camera's motion was controlled by the eye command input defined as up/down and left/right in the manual eye movement model. The maximum rotation angle is limited to 25° in 4 directions. A miniaturized and portable VNA (LibreVNA, ZeenKo) was employed to collect the SCL's signal continuously with a sample rate of 50 Hz. The 4 tags' responses at the maximum angle of the 4 directions were pre-measured to confirm the threshold for motion judgement. When the signal of the SCL met the preset threshold and lasted for 0.5 second, the corresponding eye movement was identified and eye command was transmitted to the PTZ camera wirelessly.

The thresholds set in the eye-machine interaction experiments have been supplemented in the 'Methods' section in the revised manuscript.

A deeper explanation of the "implicit swirling calibration method" has been made in the reply to Comment 5. The working procedure and validation of its accuracy also have been discussed in detail.

LabVIEW was employed for creating home-made automated program such as eye-drawing 'NJU' letters and 'snake' pattern, eye-control 'Gluttonous Snake' game, eye-web interaction, eye-controlled PTZ camera, and eye-driving robot vehicle. The LabVIEW programs implemented the procedure of signal acquisition via communication with the VNA, signal processing, real-time display, data storage, and hardware and software interaction (like web interaction, PTZ camera control, and vehicle drive). Besides, MATLAB was employed to build the response model using the thin plate interpolation in its Curve Fitting Toolbox. And Origin was employed to visualize the data. The softwares used in the experiments have been supplemented in the 'Methods' section in the revised manuscript.

References

1. Ebisawa, Y. & Fukumoto, K. Head-Free, Remote Eye-Gaze Detection System Based on Pupil-Corneal Reflection Method with Easy Calibration Using Two Stereo-Calibrated Video Cameras. *IEEE Trans. Biomed. Eng.* **60**, 2952-2960 (2013).
2. Baek, S.J., Choi, K.A., Ma, C., Kim, Y.H. & Ko, S.J. Eyeball model-based iris center localization for visible image-based eye-gaze tracking systems. *IEEE Trans. Consum. Electron.* **59**, 415-421 (2013).
3. Homayounfar, S.Z. et al. Multimodal Smart Eyewear for Longitudinal Eye Movement Tracking. *Matter* **3**, 1275-1293 (2020).
4. Robinson, D.A. A Method of Measuring Eye Movement Using a Sieral Search Coil in a Magnetic Field. *IEEE Trans. Bio-Med. Electron.* **10**, 137-145 (1963).
5. Shi, Y. et al. Eye tracking and eye expression decoding based on transparent, flexible and ultra-persistent electrostatic interface. *Nat. Commun.* **14**, 3315 (2023).
6. Frey, M., Nau, M. & Doeller, C.F. Magnetic resonance-based eye tracking using deep neural networks. *Nat. Neurosci.* **24**, 1772-1779 (2021).
7. Ku, M. et al. Smart, soft contact lens for wireless immunosensing of cortisol. *Sci. Adv.* **6**, eabb2891 (2020).
8. Kim, J. et al. A soft and transparent contact lens for the wireless quantitative monitoring of

- intr aocular pressure. *Nat. Biomed. Eng.* **5**, 772-782 (2021).
9. Zhang, J. et al. Smart soft contact lenses for continuous 24-hour monitoring of intraocular pressure in glaucoma care. *Nat. Commun.* **13**, 5518 (2022).
 10. Kim, K. et al. All-printed stretchable corneal sensor on soft contact lenses for noninvasive and painless ocular electrodiagnosis. *Nat. Commun.* **12**, 1544 (2021).

REVIEWERS' COMMENTS

Reviewer #1 (Remarks to the Author):

The authors have addressed my comments and suggestions. I believe the paper is now suitable for publication in its current version.

Reviewer #2 (Remarks to the Author):

Acceptable

Response to Reviewer 1 Comments

Comments:

The authors have addressed my comments and suggestions. I believe the paper is now suitable for publication in its current version.

Reply: Thank you very much for your valuable suggestions. And we thank the reviewer for the recommendation of publishing our manuscript in this journal.

Response to Reviewer 2 Comments

Comments:

Acceptable.

Reply: Thank you very much for your valuable suggestions. And we thank the reviewer for the recommendation of publishing our manuscript in this journal.